# Implementation of hybrid optimized battery controller and advanced power management control strategy in a renewable energy integrated DC microgrid

**Khaizaran Al sumarmad** [ID]*, **Nasri Sulaiman, Noor Izzri Abdul Wahab** [ID], **Hashim Hizam**

Faculty of Engineering, Department of Electrical and Electronic Engineering, University Putra Ma-laysia, Serdang, Selangor, Malaysia

* khaizran1977@gmail.com

**Data Availability Statement:** All relevant data are within the paper and its Supporting Information files.

## Abstract

In Renewable Energy (RE) integrated DC Microgrid (MG), the intermittency of power variation from RE sources can lead to power and voltage imbalances in the DC network and have an impact on the MG's operation in terms of reliability, power quality, and stability. In such case, a battery energy storage (BES) technology is widely used for mitigating power variation from the RE sources to get better voltage regulation and power balance in DC network. In this study, a BES based coordinated power management control strategy (PMCS) is proposed for the MG system to get effective utilization of RE sources while maintaining the MG's reliability and stability. For safe and effective utilization of BES, a battery management system (BMS) with inclusion of advanced BES control strategy is implemented. The BES control system with optimized FOPI controllers using hybrid (atom search optimization and particle swarm optimization (ASO-PSO)) optimization technique is proposed to get improved overall performance in terms of control response and voltage regulation in DC network under the random change in load profile and uncertain conditions of RE sources in real time.

## 1. Introduction

Nowadays, the deployment of DC Microgrids (MGs) is more desirable due to the widespread usage of DC-powered renewable energy (RE) sources with rapid advancement of power electronics technology, and gradual increase of DC loads in most of the energy applications [1,2]. Additionally, DC MGs are more attractive than traditional AC MGs due to a number of benefits, including higher reliability, performance, and efficiency; no reactive power and harmonics or frequency conflicts; absence of synchronization problems; reduced power loss; and less complexity in control system design [2–4]. DC MG can be operated either with grid connected or islanded (Autonomous) mode. The RE integrated DC MG operating in autonomous mode is widely used to meet the energy demand in remote areas where grid connectivity is unavailable. The RE sources such as wind and photovoltaic (PV) are used as primary energy sources in microgrids can lessen the carbon footprint and its environmental repercussions.

**Funding:** The authors received no specific funding for this work.

**Competing interests:** The authors have declared that no competing interests exist

The RE sources, on the other hand, are intermittent in nature and weather dependent. The frequent power variation caused by the intermittency of RE sources can create power and voltage imbalance in the DC power network along with challenges of maintaining reliability, stability and power quality [2,5,6]. In order to maintain better power balance (between generation and demand) and voltage regulation, the intermittency of power variation from RE sources must be mitigated or smoothed out with reliable back up power sources in DC MG. There are different energy storage (ES) devices are used as backup power source in DC MG to smooth out the variation of power from the RE sources. Mostly, for mitigating power variation in RE integrated DC MG, the battery energy storage (BES) can be considered as one of the promising ES technologies [1,2]. The power balance and DC bus voltage regulation in MG network can be maintained by allowing the BES to charge (during excess generation) and discharge (during deficit of generation) power in controlled manner [7,8].

Especially if the generators are predominantly based on RE sources and the MG is operating in autonomous mode, a BES based coordinated power management control strategy (PMCS) [2,5,7,8] is required for the MG system to get effective utilization of RE sources while maintaining the MG's reliability and stability. A variety of power management control techniques have been used by the researchers to achieve stable and reliable operation of the DC MG under various dynamic conditions. In [7] a model predictive control approach of PMCS has been proposed to manage power control among generators (PV and wind), load, and battery, as well as to stabilize the DC bus voltage of islanded DC MG under varying environmental and load conditions. A fuzzy logic-based supervised PMCS has been proposed in [9] to achieve sustainable voltage control of hybrid DC MG. The power management strategy described in [10] used BES control to regulate the dc link voltage of an islanded hybrid MG (solar PV-hydro-BES). To offset the rapid change in load power and solar PV dynamics of DC MG, authors in [11] proposed PMCS with implementation of active power control of BES. Based on DC bus signal, a decentralized storage and generation coordination technique has been proposed for DC MG [12]. Similarly, authors in [13] proposed an adaptive DC bus signaling based PMCS for a solar PV and multi-storage integrated DC MG, where the power is controlled among generation sources and storage devices with different modes of operation. To regulate DC bus voltage and manage power sharing among photovoltaics, loads, and storage devices, an ANN-based power management strategy is proposed [14]. In order to get better co-ordinated power sharing among sources and load of DC MG, an improved PMCS has been proposed with consideration of voltage and state of charge (SOC) based control strategy of BES, Fuel cell, and electrolyzer in network [15].

According to the PMCS of DC MG literature, BES can be used to get better regulation of DC bus voltage by maintaining power balance between generation and demand under normal and uncertain conditions. However, high charging/discharging ratio of BES during high intermittency of RE sources, increases stress on the battery, which can reduce its lifetime [5,8]. Furthermore, in the event of a crisis emergency (i.e., total failure of renewable sources), any mismanagement of power flow from the BES may increase reliance on conventional backup power sources (diesel/gas generators), increasing power generation costs [5,8,16]. Therefore, in order to resolve the aforementioned problems, an advanced battery management system (BMS) is needed for the batteries. To increase the availability and lifetime of BES in DC MG, the BMS is designed to provide the following features: (i) regulates BES's charging/discharging ratio, (ii) guarantees an adequate state of charge (SOC), and (iii) decreases charging stress on the battery [16].

The BMS is primarily associated with controllers of DC converters used in battery systems. Mostly, conventional integer order (proportional Integral (PI)) controllers have used for DC converters of battery systems. However, it is difficult to ensure stability of MG system and

operational performance of BES, if the conventional PI controllers are utilized in the control system of power converters while dealing with intermittent power from RE sources and load uncertainties in the MG network [17]. Because the conventional PI controllers underperforms and may not be optimal for maintaining system stability under such variable circumstances [8,18]. To improve the performance of conventional PI controllers, an alternative control concept of Fractional Order Proportional Integral (FOPI) controller can be used to ensure the robustness and stability of the system under uncertainties [19]. The FOPI controller outperforms the PI controller in terms of performance and transient response because it has one more adjustable fractional-order integral parameter [19,20].

The researchers used a variety of optimization methods to optimise the control parameters of FOPI controllers used in power converters. An improved frequency droop method (IFDM) with combination of optimized FOPI controllers have been proposed to get accurate current sharing among sources and better voltage regulation in low voltage DC (LVDC) Microgrid [21]. In a grid tied PV with hybrid energy storage system network, a chaotic grey wolf optimized FOPI controllers have been utilized for the voltage source converters to maintain DC and AC voltage regulation in network during induced dynamic conditions [22]. In fuel cell (FC) connected distributed generation system, the FOPI controller of converter has been tuned with the application of Eagle Perching Optimization (EPO) technique. The proposed EPO tuned controller provides enhanced control performance with minimized power quality (PQ) issues in the system [23]. For the application of frequency stabilization in multi-area interconnected power system, several optimization algorithms like lion algorithm technique (LAT) [24], big bang big crunch (BBBC) method [25], salp swarm algorithm (SSA) [26], and hybrid SSA with simulating annealing (SSA-ST) technique [27] have been used to tune the FOPI controllers. Likewise, a stochastic optimization technique called a sine cosine algorithm (SCA) has been adopted for optimizing FOPID controller of a hybrid energy power system (HEPS) coordinated with reheat thermal power plant. Furthermore, the quasi-oppositional based learning (Q-OBL) with adoption of SCA tuned FOPI controller can ensures the balance between power generation and load profile of system network [28]. The implementation of optimised FOPI controllers for the power converters of Distributed Generation sources (solar thermal dish-Stirling (STDS) and BESS) ensured voltage stability in an autonomous DC Microgrid. The controller gain constants are tuned using well-known meta-heuristic optimisation algorithms such as the grey wolf optimiser (GWO), mine blast algorithm (MBA), and particle swarm optimisation (PSO). The GWO optimised FOPID controller is the best option among FOPID controllers for ensuring DC bus voltage stability and profile for the studied STDS-based DC micro-grid system. [29].

The literature claims that optimised FOPI controllers outperform conventional PI controllers in terms of control response under steady state and dynamic conditions when used in power converter and power system applications. As a result, in this study, a new hybrid optimization technique using the Atom search optimization (ASO) algorithm with a PSO approach, namely ASO-PSO, is proposed to tune the control gains of the FOPI controllers used in BESS control system of DC MG. ASO is a new molecular dynamics-influenced physics-inspired optimization technique [30,31]. ASO tuned controllers have outperformed (in terms of control response under dynamic conditions) other intelligence techniques in the application of frequency control in multi-area and hybrid power systems (HPS). However, ASO, like other algorithms, has significant drawbacks, including poor exploitation, falling in local optima, and an insufficient balance of exploration and exploitation [32]. Therefore, in order to strengthen the exploration and exploitation characteristics of ASO controller, a hybridization of ASO with widely used PSO algorithms is proposed in this study. The strategy of PSO algorithm can be implemented easily and it has a simple structure. Also, PSO requires

fewer parameters and achieves faster convergence. The performance of proposed ASO-PSO optimized FOPI controllers are found to be robust and offers superior performance than ASO and PSO tuned FOPI controllers. Furthermore, in RE integrated DC MG, most of the research works was considered to validate the performance of controllers under the typical or random varying solar insolation and wind speed conditions. However, to verify the robustness of controllers, it is necessary to validate under the real time scenario. Hence, in this study, in addition to the random change of load and RE sources (solar and wind), the proposed hybrid optimized FOPI controllers of BES system was verified (in terms of control response and voltage regulation) under the real time varying solar irradiance of PV system.

The main contributions of this study are summarized as below:

- An energy storage-based power management control strategy is developed for mitigation of power variation from the Renewable Energy sources (Solar PV and Wind Turbine) to get better power balance and voltage regulation of hybrid DC Microgrid.

- A hybrid optimization technique (ASO-PSO) is adopted for the FOPI controllers of BES system to enhance the control performance in DC Microgrid under various dynamic conditions.

- The proposed robust control strategy of BES system is validated under the uncertain conditions of Renewable Energy sources in real-time.

The structure of manuscript is organised as follows: Section 2 describes the RE integrated DC MG model; Section 3 explains the power management control strategies of DC MG under various scenarios; Section 4 presents the fundamental concept of FOPI controllers; Section 5 defines the closed loop control strategy of BES converter; Section 6 details the optimization methodology, which includes the details of ASO, PSO, and hybrid ASO-PSO algorithms; Section 7 presents the results and discussion of power management control strategy and performance of optimized FOPI controllers of BES system under uncertainties of RE sources and change in load profile; and Section 8 concludes the outcomes of study with future scope of research work.

## 2. DC microgrid model

Fig 1 shows the block diagram of DC MG model which includes Main grid source, solar PV, Wind turbine, BES, and loads. The model is created using the Matlab-Simulink software environment, and it includes an optimised control system for the BES system as well as coordinated power management strategies. The main grid source (AC 400V, 10 MVA, 50 Hz) is interconnected to the DC common bus of MG through Rectifier1. The main grid source provides power support during deficit of power generation in the DC network. The wind turbine generator (12 kW rate capacity at nominal wind speed of 12 m/sec) is interconnected to the DC bus network through Rectifier 2. The solar PV unit ((i) Type: Soltech-1STH-FRL-4H-250-M60-BLK; (ii) rated capacity 8.7 kW at STC (1000 watts/m$^2$); (iii) number of cells per module: 60 (each module maximum power 248.97 watts, open circuit voltage 38. 4 V and short circuit current 8.87 Amps); (iv) number of modules per string: 4; (v) number of parallel strings: 12)) is integrated in to the DC network via DC boost converter with implementation of maximum power point tracking control algorithm (Incremental conductance) [33] to extract the maximum power from PV at given voltage conditions. In addition, a lithium-ion battery storage (220 V, 100 Ah) with bi-directional DC converter and two types of DC loads (Load 1: 8.5 kW and Load 2: 2.5 kW) are connected in the network of DC MG.

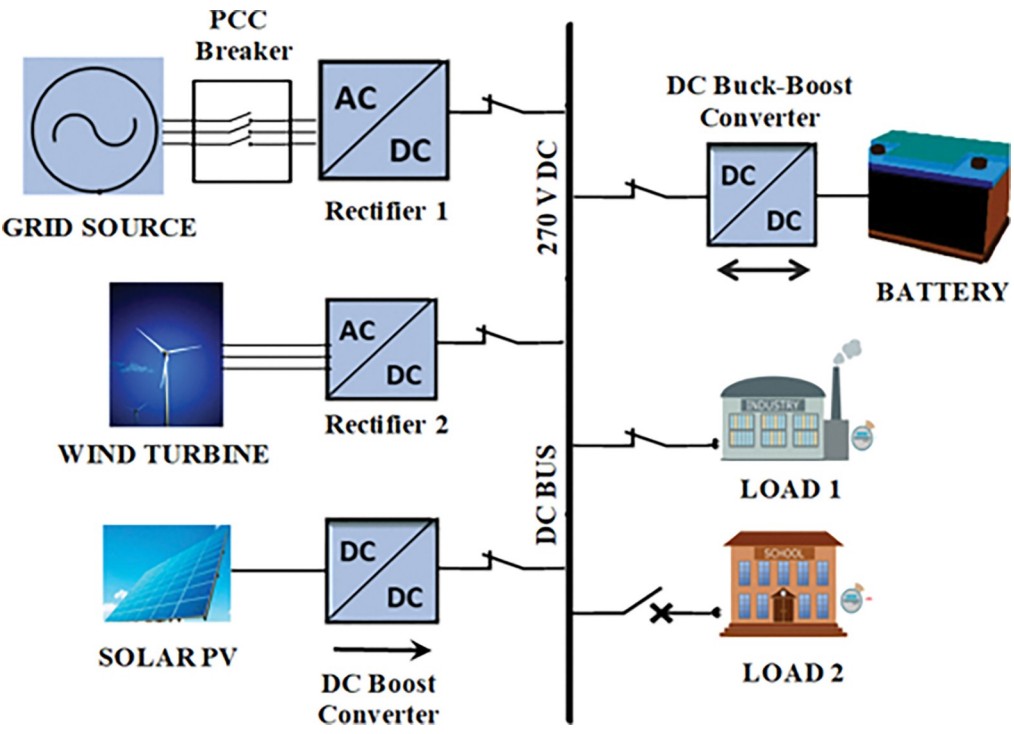

**Fig 1. DC microgrid model.**

## 3. Power management Control Strategy (PMCS)

The basic architecture of PMCS, as shown in Fig 2 ensures the power balance between generation and demand in all the time to get better voltage regulation in DC network. In addition, it ensures safe operation of BES system, effective utilization of RE sources, and enhancement of system reliability/stability. Based on the availability of power generation and existing demand in DC MG, it guides the BMS control of BES system to get appropriate charging and discharging of battery within safest operating range of state of charge (SOC). Furthermore, during the case of power deficit from RE sources and non-availability of battery power, it initiates source control to get power support from main grid source through switching (on/off) of breaker at point of common coupling (PCC).

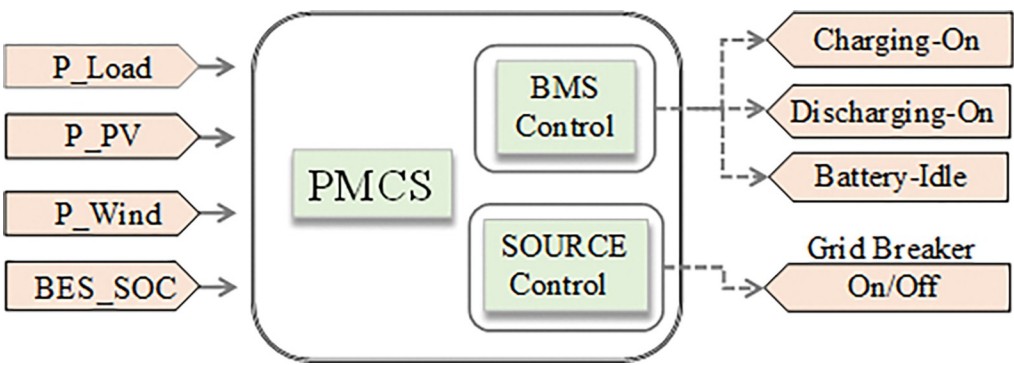

**Fig 2. Basic architecture of PMCS.**

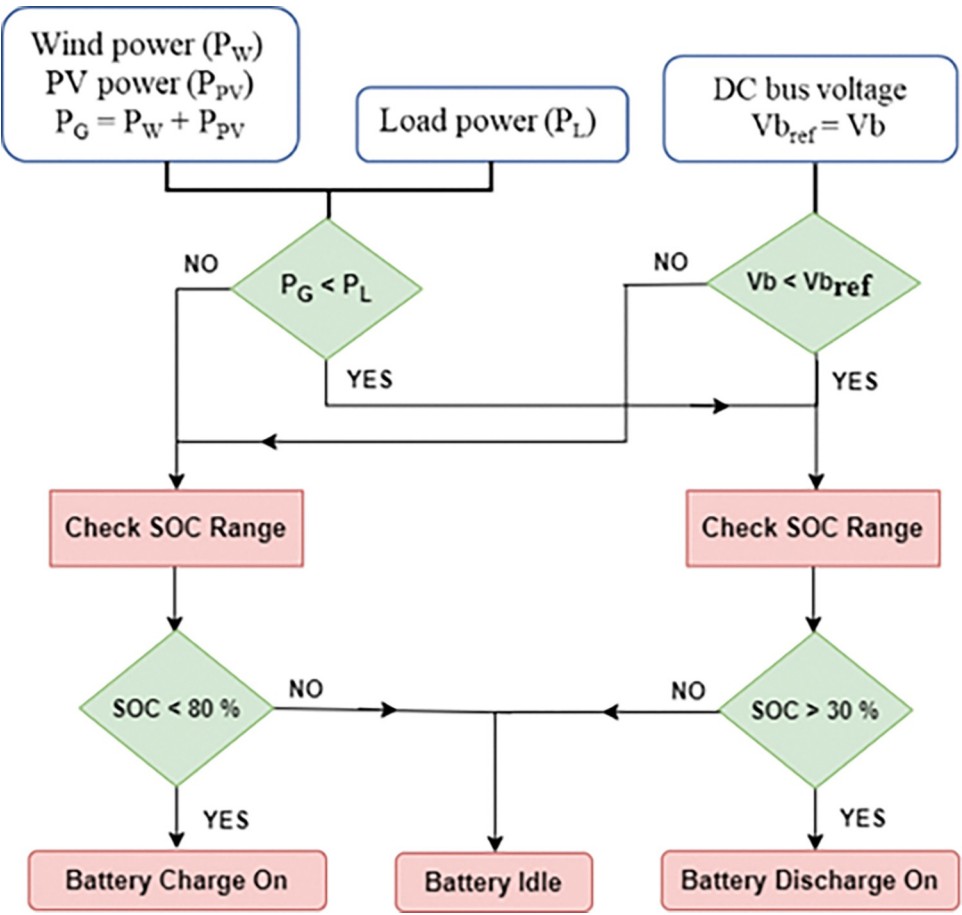

**Fig 3. PMCS with BMS control (battery charging/discharging).**

The PMCS control strategy, depicted in Fig 3, initiates the BES into charging or discharging mode based on the total power generation available in relation to total demand, as well as the difference in DC bus voltage between actual and reference values in the DC MG. The BMS control considers the battery's SOC to be in the range of 30 to 80% while battery is in operation. This range of SOC needs to be maintained during operation to protect the battery from under-discharge (below 30% of SOC) and over-charge (above 80% of SOC) conditions. When the battery's SOC is greater than 30% and the total power generation ($P_G$) in DC MG is less than the total power demand ($P_L$), the battery is allowed to discharge as necessary. On the other hand, the battery is permitted to charge appropriately if the $P_G$ in the DC network is greater than the $P_L$ and the SOC of the battery is lower than 80%. If the battery SOC falls below 30% during discharge and exceeds 80% during charge, the battery enters an idle state.

Fig 4 depicts the PMCS with source control. If total power generation $P_G$ in the DC network is less than total load $P_L$, and the battery is depleted (less than 30% SOC), the load management control initiates switching off non-essential loads (load shed) in order to achieve power balance in MG. Even after load shed, if generation is less than demand, the PMCS initiates source control to obtain power from the grid source by turning on the Grid breaker at the PCC. As a result, when the DC MG is operating in autonomous mode and the generators are primarily powered by RE sources (solar and wind), this types of proposed PMCS can be used to increase the reliability of the energy supply.

## 4. Fundamental concept of FOPI controllers

An additional non-integer integrator order that can be adjusted using an optimization technique. This additional parameter of FOPI improves the dynamic response of a system characteristic when compared to the conventional PI controller. Furthermore, by responding faster in the face of large disturbances, the FOPI controller outperforms the PI controller in terms of stability, robustness, and maintaining monotonic behaviour. These distinctive features of FOPI increase the motivation for using fractional-order controllers in the current paper [34]. The FOPI controller's transfer function is typically expressed as,

$$Tf(s) = K_{Pf} + \frac{K_{if}}{S^{\lambda}} \tag{1}$$

According to Eq (1), three parameters ($K_{Pf}$, $K_{if}$, $\lambda$) must be appropriately designed using control optimization techniques. The proportional and integral gains of the controller are denoted by $K_{Pf}$ and $K_{if}$, respectively, and $\lambda$ denotes the order of the integrator. Taking into account the integral time function, Eq (1) can also be written as [19],

$$Tf(s) = K_{Pf}(1 + \frac{K_{if}}{T_{if}S^{\lambda}}) \tag{2}$$

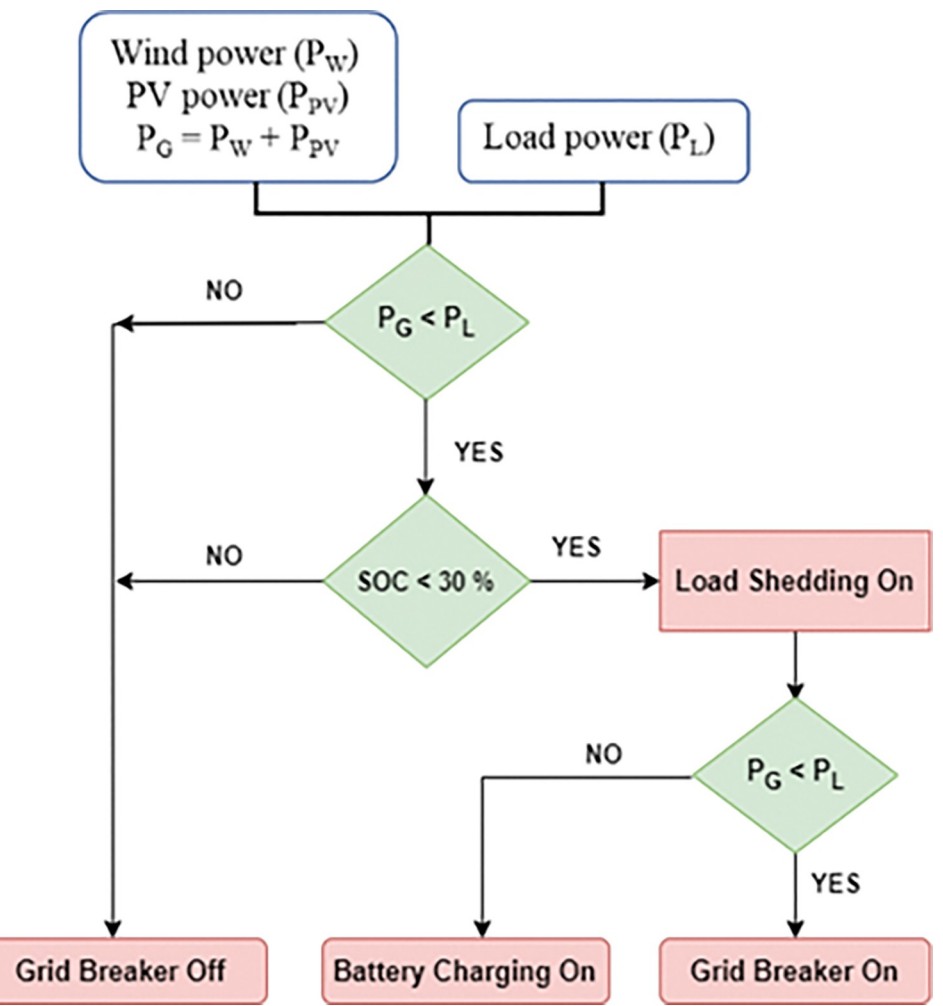

**Fig 4. PMCS with source control (Grid source on/off).**

where $T_{if}$ is the integral time function with consideration of frequency domain (jω) and the Eq (2) can be expressed as,

$$\text{Tf}(j\omega) = K_{Pf}\left(1 + \left(\frac{\text{Cos}(\frac{\lambda\pi}{2})}{T_i\omega^\lambda}\right) - j\left(\frac{\text{Sin}(\frac{\lambda\pi}{2})}{T_i\omega^\lambda}\right)\right) \tag{3}$$

In this study, the FOPI controller is implemented in BES control of DC MG, using MATLAB environment with FOMCON toolbox. The control gains of FOPI controllers are tuned through different optimization techniques such as ASO, PSO, and proposed ASO-PSO.

## 5. Closed loop control strategy with DC converter of BES

In this paper, a hybrid ASO-PSO optimised FOPI controller for the cascaded control loops of the BES is proposed to minimize the DC bus voltage transients and improve the control performance of BMS control system, under the dynamic conditions of MG network. Fig 5 depicts the closed loop control system of the DC bi-directional converter used in the BES system. The voltage control loop regulates the MG network's DC bus voltage and provides a reference current signal to the inner current loop. Based on the computing error between the actual battery current (Ib) and reference current (Ib-ref) signals, the inner FOPI controller generates an output control signal.

The PWM generator's gating signal switches the converter based on a comparison of the control signal of the current loop and the carrier signal. In this study, the integral time absolute error (IAE) as given in Eq (4) is considered as an objective function (J) while optimising the control gains (KP, Ki, and λ (voltage controller); KP1, Ki1, and λ1 (current controller)) of FOPI controllers used in the BES control system. The objective function (J) is defined as follows:

$$\text{Minimize J} = \text{IAE} = \int [|e(t)|]dt \tag{4}$$

Where, e denotes the error signal of DC bus voltage

## 6. Methodology of optimization techniques

In this study, the FOPI controllers of the BES system are optimised using various optimization techniques. The optimization techniques ASO, PSO, and the proposed hybrid ASO-PSO are explained below:

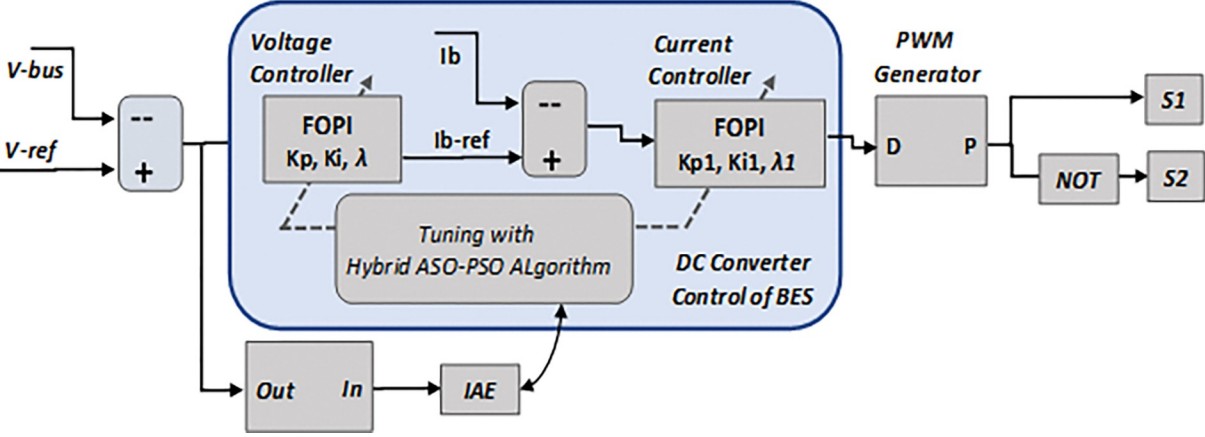

**Fig 5. Closed loop control strategy with DC converter of BES.**

## 6.1 Atom Search Optimization (ASO) approachs

The molecular dynamics theory, which offers a microscopic view of the structure of matter and how atoms interact, served as the basis for the development of the new population-based metaheuristic algorithm known as ASO. ASO uses the interaction forces, such as the force between two atoms and the attraction and repulsion from potential energy, to conduct a search. When atoms are crowded together closely, there is a force called compression that repels them. When atoms move over a wider separation, the attraction force comes into play. Any two atoms are held together by covalent bonds thanks to a constraint force. During the iterations, these forces between the atoms cause them to interact with one another. The interaction force is characterised by attraction when the distance between two atoms is less than the equilibration distance; it is characterised by repulsion when the distance is greater than the equilibration distance. Throughout the entire iterative process, the constraint force between each atom and the atom with the best fitness remains constant [30,31]. Refer [19] for a more thorough explanation of ASO.

## 6.2 Particle Swarm Optimization (PSO) approach

PSO is a population-based stochastic optimization with flocking bird behaviour. Each solution in PSO is symbolised by a "bird" in the particle-based search space. The first step in using PSO to solve an optimization problem is to randomly initialise a predetermined number of particles in the search space. To update the fitness value, every particle in the search space has a specific velocity and position. The best solution found across the entire search space is defined as the global best, and the particles record their personal best during each iteration. In order to update the particle's new position, each particle's velocity is updated based on its personal best and global best values. When the global best fit value reaches a predetermined tolerance or the number of optimization iterations reaches a maximum value, the algorithm stops. Refer [35] for a more thorough explanation of PSO.

## 6.3 Hybrid ASO-PSO optimization approach

Numerous meta-heuristic algorithms have demonstrated their suitability for solving complex and challenging real-world optimization problems across a variety of industries. Enhancing the ability of meta-heuristics to find more precise solutions quickly becomes increasingly important as optimization problems get bigger. The achievement of a balance between exploration and exploitation is another, more important task. While exploitation looks for potential solutions that have already been found, exploration searches the entire search space for promising regions. Although slow in convergence, algorithms with excellent exploration ability rarely trap in local optima. While they are susceptible to local optima, algorithms with high exploitation capability, on the other hand, move the search closer to the ideal location. To get around this, the author suggests a brand-new hybrid technique that combines exploration and exploitation features to produce the best solution with the least amount of computational effort.

ASO is suitable for many real-world optimization problems and, as was previously mentioned, was inspired by molecular dynamics [30]. Atoms' velocities, which are governed by their masses, constraint forces, and interaction forces, govern how they move throughout the entire iteration process. The convergence rate degrades and traps in local optima, resulting in poor convergence because there is no global best updating in ASO [32]. This issue was solved by combining PSO's global search capability with hybridization in order to balance both exploration and exploitation. However, PSO, a probabilistic variation of the global search optimization algorithm, has been used in numerous engineering applications. The search space is

traversed by each particle in PSO at a variable velocity that is dynamically adjusted based on its and other particles' flight experiences. In PSO, each particle aspires to evolve by stealing traits from its prosperous neighbours. Each particle also has a memory, allowing it to recall the ideal location it has ever found while searching. The following equation is used to calculate the particle's updating velocity [35].

$$u_i(t+1) = \omega.u_i(t) + c_{o1}.r_{a1}.(l_{best,i} - c_i(t)) + c_{o2}.r_{a2}.(gl_{best} - c_i(t)) \tag{5}$$

where $c_i(t)$ and $u_i(t)$ are the $i^{th}$ particle's current position and velocity at the $i^{th}$ iteration, respectively, and $l_{best}$ and $gl_{best}$ are the $i^{th}$ particle's local and global best positions, respectively. The $c_{o1}$ and $c_{o2}$ are the social and cognitive coefficients, $ra_1$ and $ra_2$ are the distinct random numbers with a range of 0 to 1, and $\omega$ is the inertia weight. Due to the memory set that the algorithm maintains, PSO has the advantages of good exploitation and convergence ability. In order to minimize the voltage error in the DC bus of MG network, an AS-PSO based hybrid strategy is proposed to optimize the gains of FOPI controllers used in DC converter of BES.

The steps for optimizing the gains of BES FOPI controllers using the ASO-PSO algorithm are as follows:

**Step 1:** First, assign the ASO-PSO parameters such as total number of atoms ($N_a$), multiplier weight ($\beta_m$), number of iterations ($T_i$) and depth weight ($\alpha_w$), etc. Initialize the controller's parameters at random and specify their positions as $C_i = (c_i^1, \ldots c_i^{dn}, \ldots c_i^k)$; i = 1, 2…., k, where i and dn specifies the atom and the dimension of the search region (dn = 1, 2, … Dn) in a Dn dimension region.

**Step 2:** Next, determine each atom's fitness function in accordance with Eq (4)

**Step 3:** The mass of $i^{th}$ atom, $ma_i(t)$ is calculated as:

$$ma_i(t) = \frac{Ma_i(t)}{\sum_{j=1}^{k} Ma_i(t)} \tag{6}$$

$$Ma_i(t) = e^{-\frac{Fitness_i(t) - Fitness_{best}(t)}{Fitness_{worst}(t) - Fitness_{best}(t)}} \tag{7}$$

where the $i^{th}$ atom's normalized mass and mass at the $i^{th}$ iteration are represented by $ma_i(t)$ and $Ma_i(t)$. The atom's best and worst fitness values are represented by $Fitness_{best}(t)$ and $Fitness_{worst}(t)$, respectively.

**Step 4:** The interaction force $F_i^{df}(t)$ provided by other atoms on $i^{th}$ atom can be represented as:

$$F_i^{df}(t) = \sum_{j \in Kbest} rand_j F_{ij}^{df}(t) \tag{8}$$

**Step 5:** $G_i^{df}(t)$, a constraint force, is calculated as follows:

$$G_i^{df}(t) = \lambda_a(t)(c_{best}^d(t) - c_i^d(t)) \tag{9}$$

where $\lambda_a(t) = \beta_a e^{\frac{-20t}{T}}$ is the Lagrangian multiplier, the best atom found at the $t^{th}$ iteration in the $d^{th}$ position and $c_i^d(t)$ denotes the best atom attained at the $t^{th}$ iteration in the $i^{th}$ position. Through all iterations, the resultant force $F_{res}$ of the $i^t$ atom at time t can be expressed as:

$$F_{res} = F1_i + G1_i \tag{10}$$

**Step 6:** Newton's law of motion defines the acceleration of the $i^{th}$ atom as:

$$ac_i^d(t) = \frac{F1_i^d(t)}{ma_i^d(t)} + \frac{G1_i^d(t)}{ma_i^d(t)} = -\alpha\left(1 - \frac{t-1}{T}\right)^3 e^{\frac{-20t}{T}}$$

$$\times \sum_{j \in K_{best}} \frac{rand_j[2 \times (h1_{ij}(t))^{13} - h1_{ij}^7]}{ma_i(t)}$$

$$\frac{(c_j^d(t) - c_i^d(t))}{\|c_i(t), c_j(t)\|_2} + \beta_a e^{\frac{-20t}{T}} \frac{c_{best}^d(t) - c_i^d(t)}{ma_i(t)}$$

(11)

Where K is defined as $K(t) = N - (N-2) \times \sqrt{\frac{t}{T}}$. $K_{best}$ is a subset of an atom population consisting of the first K atoms with the highest fitness value.

**Step 7:** The velocity of each atom is updated by combining ASO's local and PSO's global search capabilities. This can be mathematically represented as.

$$u_i^d(t+1) = \omega \cdot u_i^d(t) + c_{o1} \cdot a_i^d(t) + c_{o2} \cdot (r_{a1} \cdot (l_{best,i} - c_i^d(t)) + r_{a2} \cdot (gl_{best} - c_i^d(t)))$$ (12)

where $r_{a1}$ and $r_{a2}$ are the random numbers between 0 and 1, $\omega$ is inertia weight linearly decreasing from 1.2 to 0.4, and $co_1$ and $co_2$ are acceleration coefficients with values 0.7 and 0.4, $gl_{best}$ denotes the particle's global best position thus far, while $l_{best}$ denotes the $i^{th}$ atom's local best location.

In (12), first half of the equation $(u_i^d(t) + c_{o1} \cdot ac_i^d(t))$ depicts the ASO velocity updating equation, which is primarily dedicated to exploration; and the second half $(c_{o2} \cdot (r_{a1} \cdot (p_{best,i} - c_i^d(t)) + r_{a2} \cdot (gl_{best} - c_i^d(t))))$ is similar to PSO, which is used to attract the atoms in the direction of resultant force, which emphasises exploitation.

**Step 8:** Once the atoms velocity is updated, each atom position can be updated as:

$$c_i^d(t+1) = c_i^d(t) + u_i^d(t+1)$$ (13)

**Step 9:** Once the required number of iterations has been reached, the best controller parameters, glbest, are obtained. If not, continue until the terminating condition is met by going through steps 2 through 8.

The proposed ASO-PSO technique is used to optimize the control parameters of FOPI controllers by minimizing the voltage error of DC network of MG system with both exploration and exploitation characteristics to reach best optimal solution. The parameters initialized for ASO-PSO are $N_a$ = 50, $T_i$ = 100, $\alpha_w$ = 50, and $\beta_m$ = 0.2. Fig 6 depicts a flowchart representing the process of minimizing the fitness function using ASO-PSO. The tuned control parameters of FOPI controllers by the PSO, ASO, and proposed hybrid ASO-PSO techniques are given as follows: (i) Control gains of PSO controllers: KP = 0.005, Ki = 0.1, λ = 0.01 and KP1 = 0.2, Ki1 = 20, and λ1 = 0.04; (ii) Control gains of ASO Controllers: KP = 0.005, Ki = 7.2, λ = 0.5 and KP1 = 0.22, Ki1 = 50, and λ1 = 0.5; and (iii) Control gains of hybrid ASO-PSO: KP = 0.005, Ki = 10, λ = 0.6 and KP1 = 0.25, Ki1 = 25, and λ1 = 0.4.

## 7. Results and discussion

The results of the simulation analysis with the implementation of the proposed PMCS in the DC MG at various scenarios, as well as the optimized outcomes of the proposed control strategy was implemented with the FOPI controllers of the BES system, are discussed in this section.

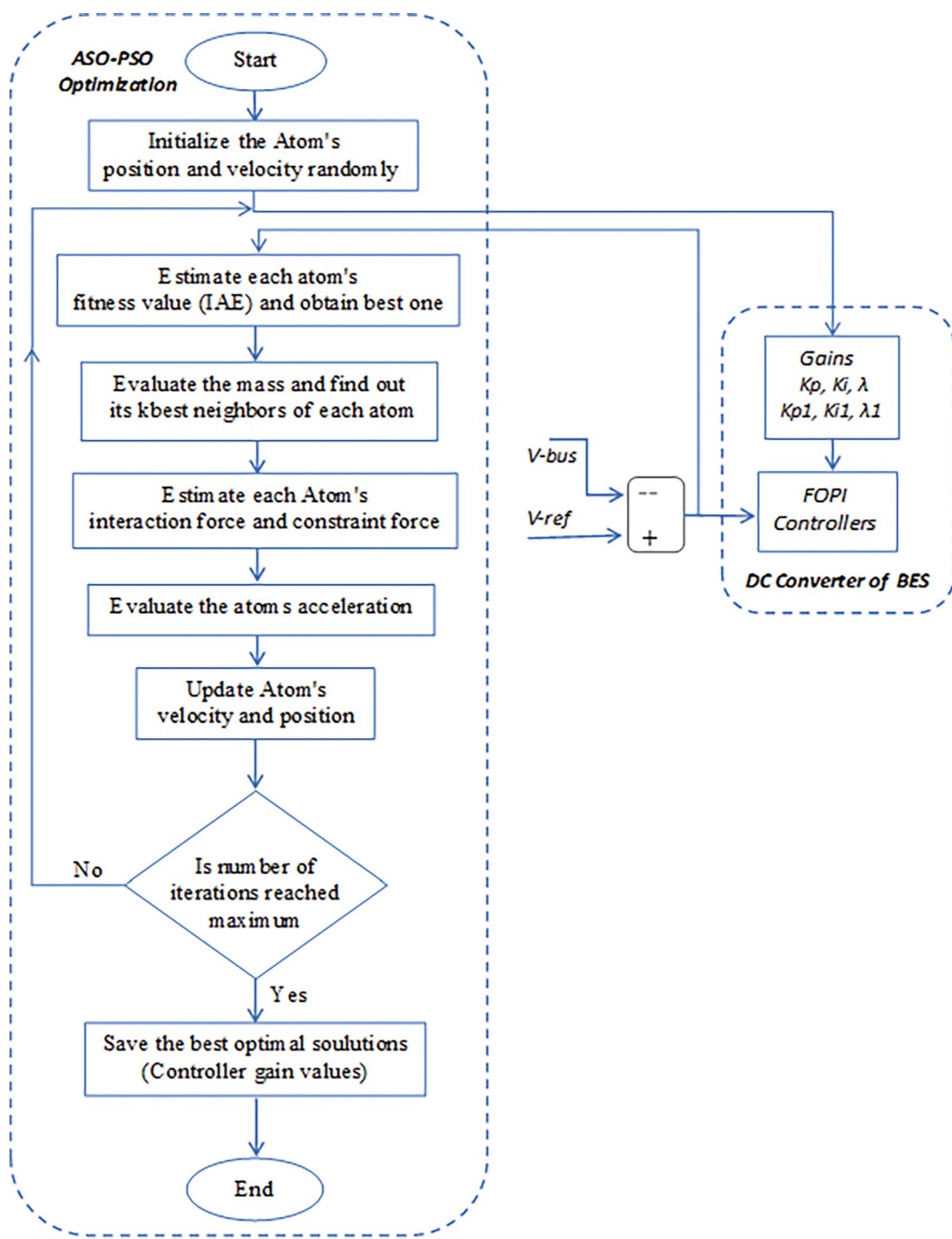

**Fig 6. Process steps for tuning of FOPI Controller swith ASO-PSO.**

## 7.1 Simulation results of proposed PMCS with BMS control

In DC MG, in order to get better voltage regulation, it is necessary to maintain power balance in all the time. The total power generation from the sources should be equal to the total demand in network. The total power generation ($P_G$) which is expressed in the Eq (14), the

battery power ($P_B$) with "+" sign represents that battery is in charging mode during excess power generation of RE sources.

$$P_G = P_{PV} + P_W + P_B \tag{14}$$

where $P_{PV}$ is the power from solar PV and $P_W$ is the power from wind turbine generator. The total power generation ($P_G$) as expressed in Eq (15), the battery power ($P_B$) with "—" sign indicates that the battery is in discharging mode during deficit power generation of RE sources.

$$P_G = P_{PV} + P_W - P_B \tag{15}$$

Total demand $P_L$ is the sum of individual loads in network, can be expressed as,

$$P_L = P_{L1} + P_{L2} + \dots P_{LN} \tag{16}$$

where $P_{LN}$ is the N-number of loads in network
Power balance in DC MG can be expressed as,

$$P_G = P_L + P_{LOSS} \tag{17}$$

$$P_{LOSS} = I^2R \tag{18}$$

where $P_{LOSS}$ is the power losses in network. $P_{LOSS}$ is generally expressed as product of network resistance (R) and square of the current ($I^2$) flowing through the DC network.

The effectiveness of proposed PMCS was evaluated under the following conditions: random varying power from solar PV (Fig 7), wind turbine (Fig 8), and step load (Fig 9A) in autonomous mode of MG operation. The PMCS ensures appropriate charge and discharge operation of battery through BMS control, in order to maintain power balance and voltage regulation in the MG network, based on the availability of power generation from RE sources and connected load in the MG network. The total period of simulation was considered around 5s and the results of battery power, SOC, and DC bus voltage are depicted in the Fig 7B–7D, respectively.

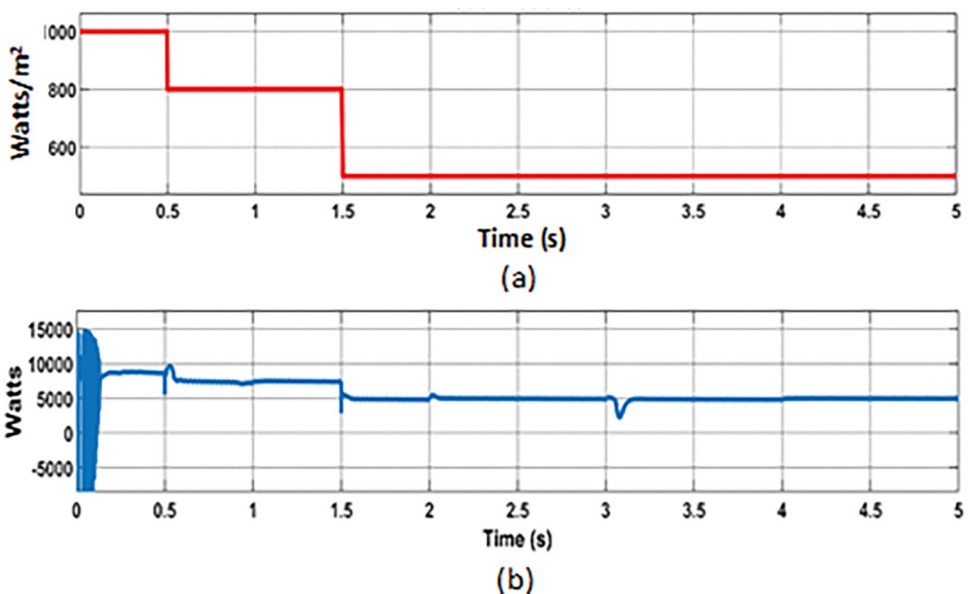

**Fig 7.** PMCS analysis with BMS control: (a) Solar irradiance; (b) PV power.

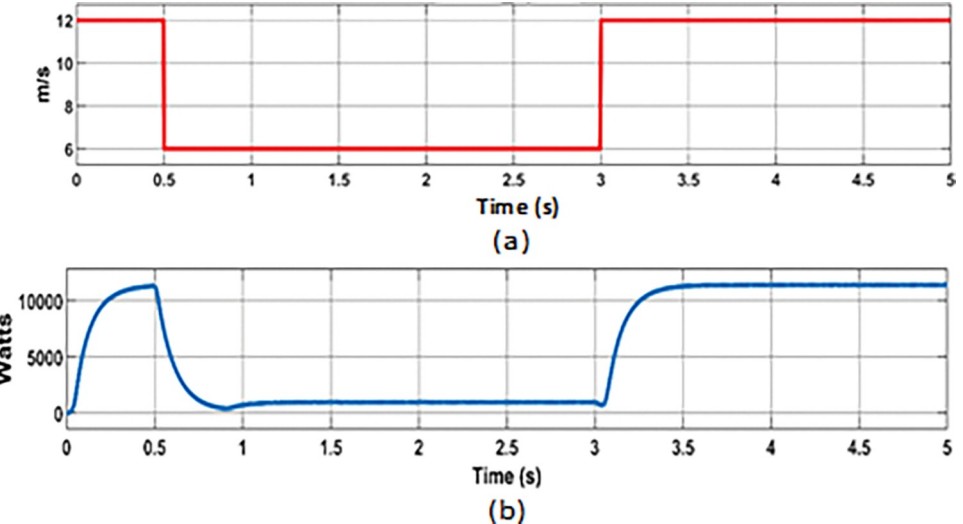

**Fig 8.** PMCS analysis with BMS control: (a) Wind speed; (b) Wind power.

In addition, the results of power balance achieved during the cases of solar, wind, and load power are depicted in Fig 10.

i. **Battery charging with excess power**

From 0 to 0.5s, maximum PV power (8.7 kW with 1000 watts/m$^2$ of solar irradiance), wind power (11.3 kW with 12m/s of wind speed) and connected load capacity of 8.7 kW was considered. During this period, the total power generation $P_G$ was higher than $P_L$ and the battery was charged with excess power (-10.625 kW). Due to charging mode, the SOC of battery was increased from 49.90% to 50.005%.

i. **Battery discharging with minimum power**

From 0.5s to 1.5s, reduced PV power (7.5 kW) was considered due to varying solar irradiance from 1000 watts/m$^2$ to 800 watts/m$^2$. Likewise, from 0.5s to 3s, reduced wind power (1 kW) was considered due to reduced wind speed from 12 m/s to 6 m/s. During this period, the connected load was considered around 8.7 kW and the total generation was almost equal to total demand. However, to meet the power losses in the MG, the battery was allowed to discharge to a minimum level of 1 kW with change in SOC (from 50.005% to 50.003%).

i. **Battery discharging with maximum power**

From 1.5s to 3s, the solar irradiance was reduced in further level (from 800 watts/m$^2$ to 500 watts/m$^2$) with corresponding reduction of PV power around 5 kW, wind power was at minimum of 1 kW (until 3s), and load was increased (at 2s) from 8.7 kW to 11 kW. During this period, due to deficit power generation (from RE sources) and increased demand in MG network, the battery was allowed to discharge maximum of around 5.3 kW with change in SOC (from 50.003% to 49.99%).

i. **Battery charging with minimum power**

From 3s to 5s, the wind speed was increased to nominal level (12 m/s), so that the wind power was increased to 11.3 kW. During this period of analysis, minimum PV power (5 kW) and maximum load (11 kW) was considered. Due to increased wind power, the battery was

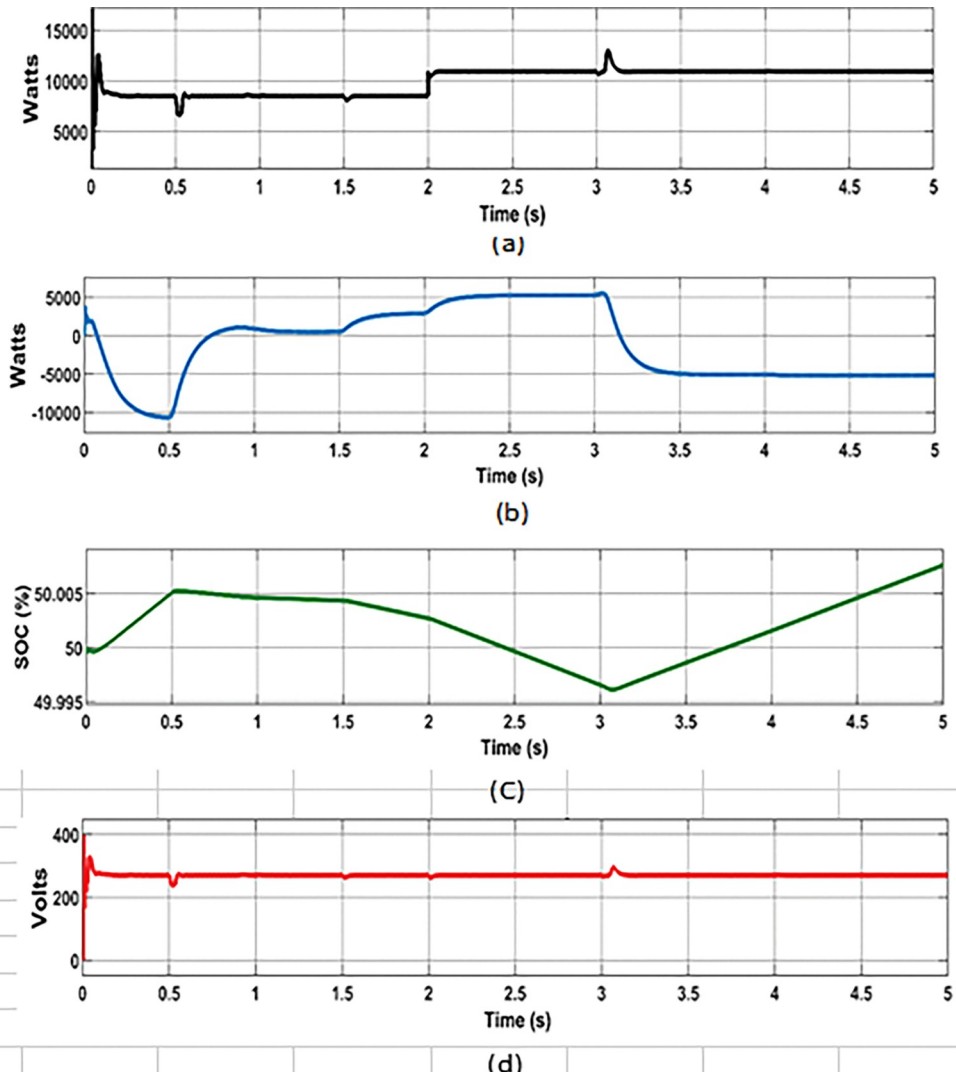

**Fig 9.** PMCS analysis with BMS control: (a) Load power; (b) Battery power; (c) Battery SOC; (d) DC bus voltage.

allowed to charge at maximum power (-5 kW) and the change in SOC level was observed (from 49.99% to 50.008%).

Furthermore, throughout the analysis, the DC bus voltage was maintained constant (270 V), even with varying power from RE sources and change in load conditions. Thus, the proposed PMCS with BMS control was more effective to maintain power balance and voltage regulation in DC network, even at uncertain conditions of RE sources and load variation.

## 7.2 Simulation results of proposed PMCS with Source control

This PMCS was developed in such a way to improve the reliability of energy supply in RE integrated DC MG under the conditions of depleted battery storage (SOC below 30%) and deficit of generation from RE sources. For this analysis, a random varying PV power (Fig 11A and 11B) and wind power (Fig 12A and 12B) with respect to the SOC level of battery storage was considered while evaluating the performance of PMCS. The results of change in load power, battery power, SOC, grid power, and DC bus voltage are shown in Fig 13A to 13E, respectively.

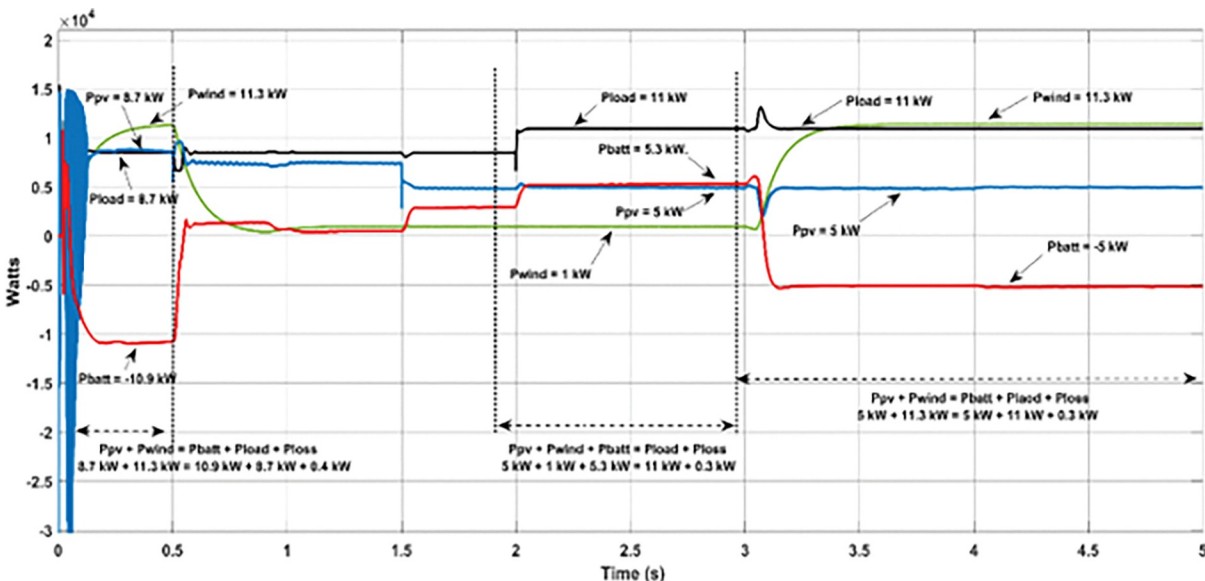

**Fig 10. Results of power balance in MG: Proposed PMCS with BES control.**

i. **Battery charging at maximum power generation**

From 0s to 0.5s, maximum power from PV (8.7 kW (at 1000 watts/m$^2$ solar)) and wind turbine (11.4 kW (at 12 m/s wind speed)) was observed at connected load capacity of 11.54 kW in DC MG. During this period, the battery was allowed to charge with excess power (-7.75 kW) and SOC was increased from 30.004% to 30.01%. From 0.5s to 1.5s, the PV power was decreased to 7.2 kW due to reduced solar irradiance (800 watts/m$^2$ solar). From 0.5s to 1s, the wind power was same (11.4 kW with 12 m/s wind speed) and after 1s, the wind power was decreased to 1 kW due to reduced wind speed (6 m/s). During this period, with same load

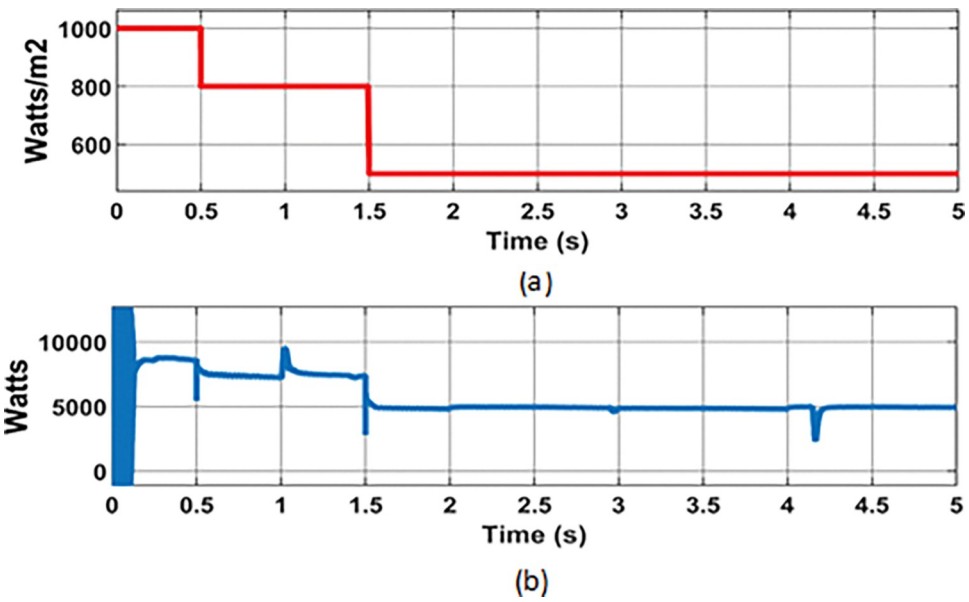

**Fig 11. PMCS analysis with Source control: (a) Solar irradiance; (b) PV power.**

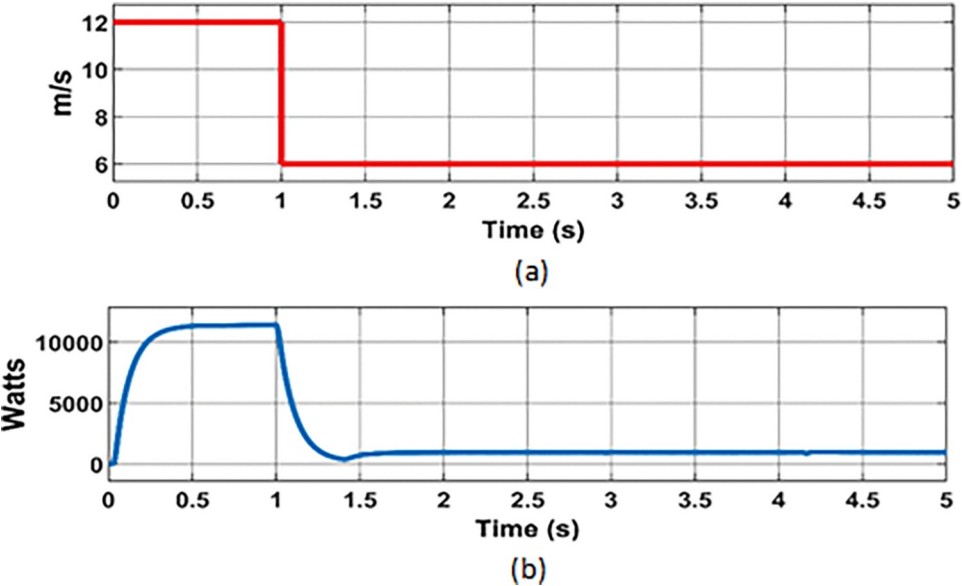

**Fig 12.** PMCS analysis with Source control: (a) Wind speed; (b) Wind power.

(11.54 kW) connected., the excess power (- 6.75 kW) was used to charge the battery with change in SOC level (from 30.01% to 30.0125%).

### i. Battery Discharging when deficit of power generation

From 1s to 1.5s, the PV power was 7.2 kW at 800 watts/m$^2$ of solar irradiance and wind power was minimum of 1 kW at 6 m/s with same connected load capacity of 11.54 kW. During that period, the shortage of power (4 kW) was discharged from battery with change in SOC level (from 30.0125% to 30.01%).

### i. Battery discharging with minimum SOC level (30%)

From 1.5s to 3s, the power generation was minimum from PV (5 kW at solar irradiance of 500 Watts/m$^2$) and wind turbine (1 kW at 6 m/s wind speed) due to reduced solar and wind speed. At that time, the battery was allowed to discharge power (5.8 kW) in further level with change in SOC level of 30%. Once the battery SOC level was reached to minimum operating level (30%), the PMCS initiates load management control to switch off non-essential loads (load reduced from 11.4 kW to 9.10 kW) in MG network.

### i. Grid source on when SOC level of battery below 30%

From 3s to 5s, even after switching off non-essential loads, the power generation from both PV and wind turbine was minimum and not sufficient to fulfil the demand, and battery was allowed to discharge in further level. Once the battery SOC level was reached below 30% (29.995% at 4.16 s) and battery was entered in idle state. At the same time, the PMCS control initiates source control to switch on Grid breaker at PCC. So that, the power support (8.35 kW) from Grid source was ensured. Once the power started to import from grid source (at 4.16 s), that battery was allowed to charge again (- 4.8 kW) and its SOC was increased from 29.995% to 30.01%.

Furthermore, the DC bus voltage was maintained at constant level (270 V) throughout the analysis even at change in power of RE sources. Thus, the proposed PMCS with source control ensures the reliability of energy supply in DC MG, even with depleted condition of battery storage and loss of power from RE sources.

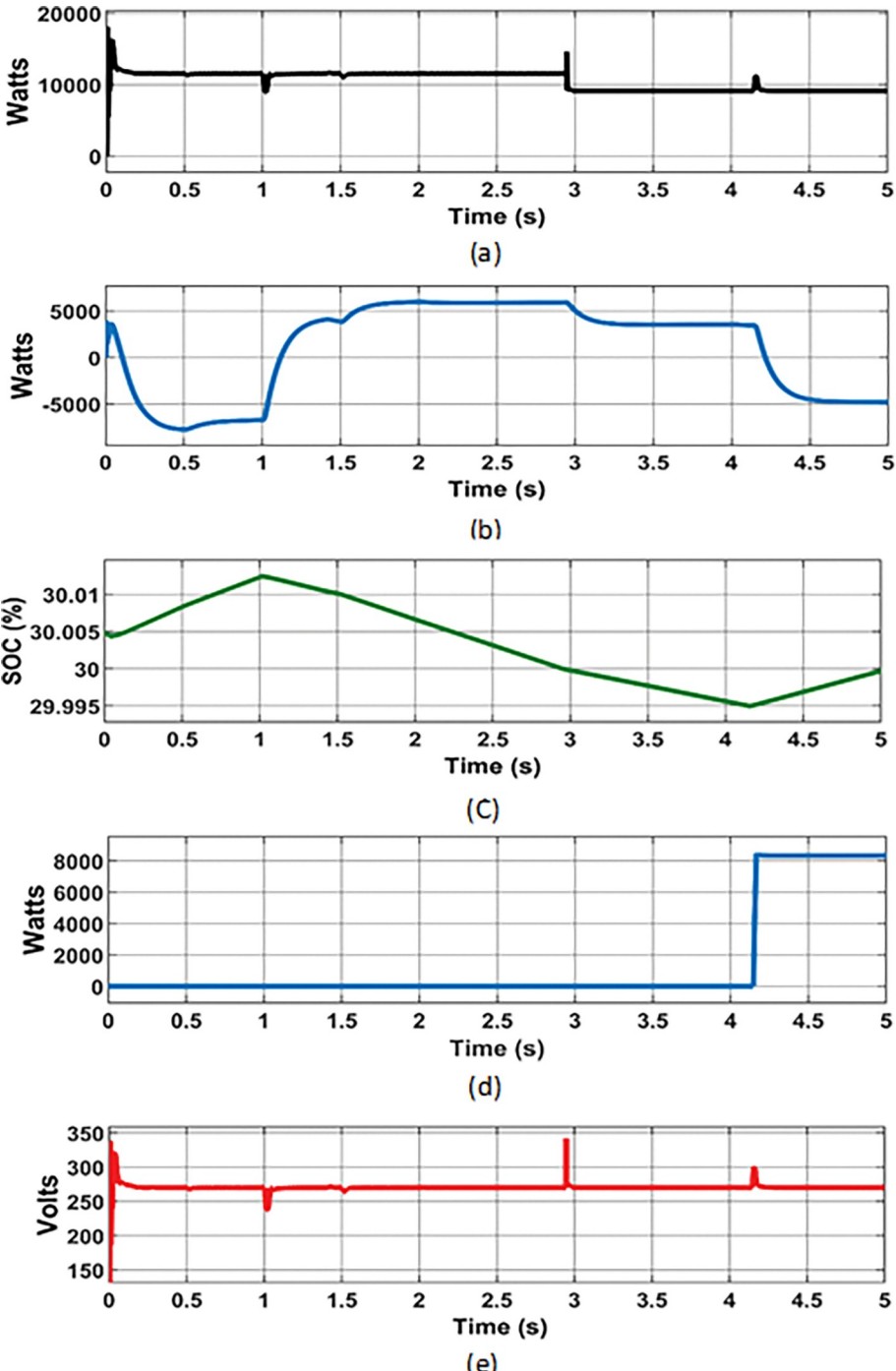

**Fig 13.** PMCS analysis with Source control: (a) Load power; (b) Battery power; (c) Battery SOC; (d) Grid power; (e) DC bus voltage.

## 7.3 Results of optimization

In this analysis, the effectiveness of FOPI controllers of BES was verified under the uncertain conditions of RE sources and connected load in MG network. The FOPI controllers with optimization results of hybrid ASO-PSO technique was compared with other ASO and PSO

optimized controllers in terms of control response (settling time, over shoot, and rise time) and voltage regulation in MG network.

**7.3.1 Analysis with random change of RE sources and load.** In this analysis, the random change of wind, solar, and load were considered over the total simulation time of 2 s as follows: (i) The change in wind speed (12 m/s to 6 m/s) at 0.5 s, (ii) change in solar irradiance (1000 watts/m$^2$ to 300 watts/m$^2$) at 1 s, and (iii) change in load (8.57 Ω to 6.67 Ω) at 1.5 s. The DC bus voltage levels (Fig 14D), during the corresponding change in wind power (11.3 kW to 1 kW (Fig 14A), PV power (8.7 kW to 3 kW (Fig 14B) and load power (8.5 kW to 11 kW (Fig 14C) were observed with optimized hybrid PSO-ASO, ASO, and PSO FOPI controllers of BES system. The results of voltage deviations (Fig 15), obtained with different optimization techniques are listed in Table 1.

According to the results of voltage deviations, as illustrated in Fig 15 and Table 1, the voltage deviations were significantly reduced with proposed hybrid ASO-PSO optimized FOPI

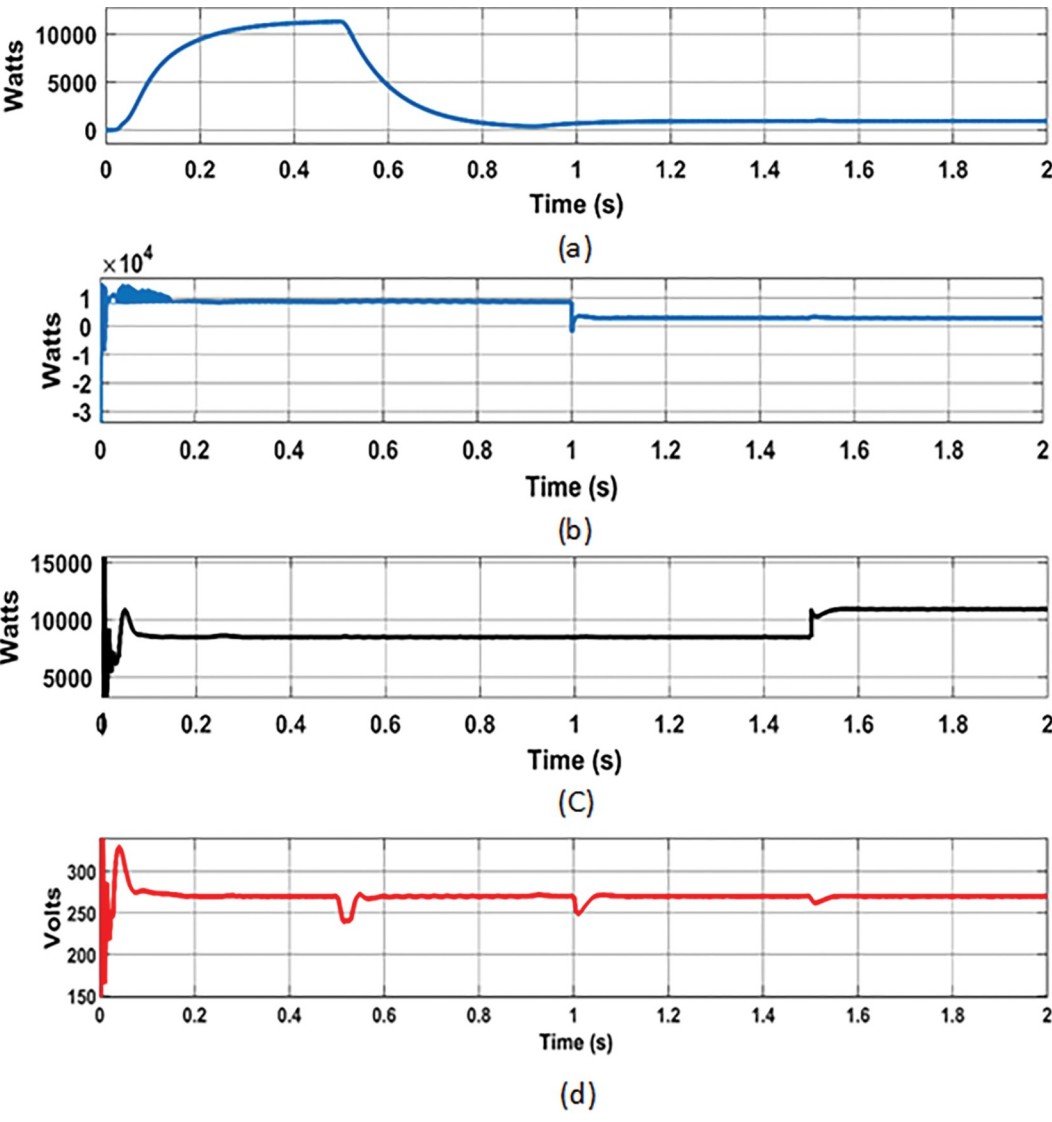

**Fig 14.** DC bus voltage analysis: (a) Wind power; (b) PV power; (c) Load power; (d) DC bus voltage.

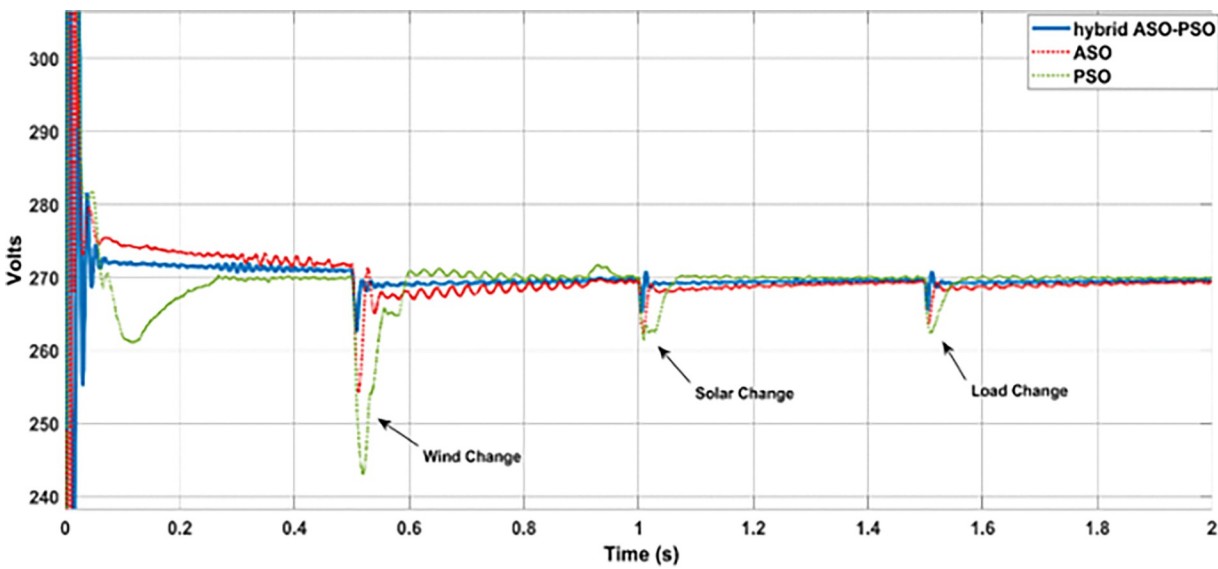

**Fig 15. Results of voltage deviations at different events.**

controllers than with other (ASO and PSO tuned) FOPI controllers. With respect to the reference bus voltage of 270 V, reduced voltage deviations were observed at different events with proposed hybrid method (2.7% (wind change), 1.7% (solar change), and 1.6% (load change)) as compared to ASO (5.8% (wind change), 2.8% (solar change), and 2.4% (load change)), and PSO (10% (wind change), 3.2% (solar change), and 2.8% (load change)) methods. Thus, as compared to ASO and PSO controllers, the proposed hybrid controller offers improved control in terms of voltage deviations (3.1% and 7.3% more than with ASO and PSO (wind change), 2.8% and 1.1% more than with ASO and PSO (solar change), and 0.8% and 1.2% more than ASO and with PSO (load change)) during the different dynamic conditions of DC MG.

Fig 16 depicts the charging and discharging power operation of the BES system under various dynamic conditions. From 0 to 1s, when the power generation from PV and wind turbine generator was maximum, the battery was allowed to charging the excess power of 10.9 kW. Due to reduced wind speed at 0.5s (wind power reduced from 11.3 kW to 1 kW), the battery was allowed to discharge power to a minimum of 0.15 kW from 0.5 s to 1s. Due to reduced solar irradiance at 1s (PV power reduced from 8.7 kW to 3 kW), the power discharge from BES was increased (0.15 kW to 4.7 kW) in further level from 1s to 1.5s. During 1.5s to 2s period, maximum power (7.25 kW) discharge from BES was allowed when the additional load (from 8.5 kW to 11 kW) was switched on at 1.5s. According to the results of power contribution from different controllers of BES system, as shown in Fig 16, the proposed hybrid ASO-PSO controller of BES takes less time to reach steady state power level (during charging

**Table 1. Results of voltage deviations.**

| Optimization Methods | DC bus Voltage Deviations | | | |
|---|---|---|---|---|
| | Wind change | Solar change | Load change | Voltage deviations |
| Hybrid ASO-PSO | 262.4 V | 265.2V | 265.4 V | - 1.7% |
| ASO | 254.2 V | 262.3 V | 263.5 V | - 2.4% |
| PSO | 242.8 V | 261.3 V | 262.4 V | - 2.81% |

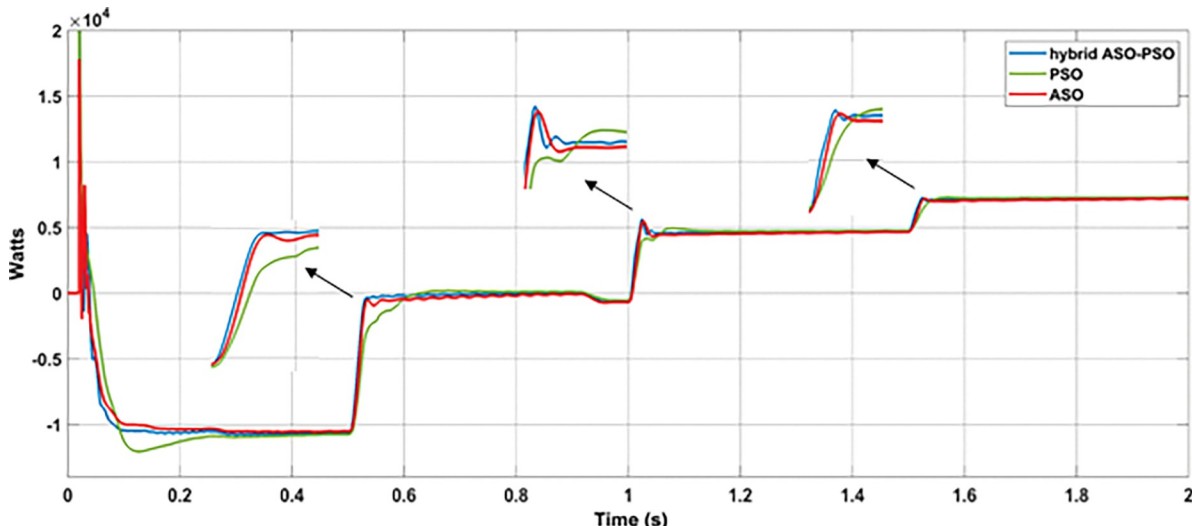

**Fig 16. Results of BES power under the random change of RE sources and load.**

and discharging operation) than other controllers (ASO and PSO) of BES. Due to the fast control response of hybrid controller, the deviations in power and voltage level were significantly reduced during the uncertain power variations from RE sources and connected load in MG network.

Furthermore, the control response of proposed hybrid ASO-PSO tuned FOPI controllers was verified in terms of settling time, rise time, and overshoot with DC bus voltage of DC MG. Fig 17 depicts the results of settling time, while Table 2 depicts the results of peak overshoot, settling time, and rise time of the proposed hybrid and other optimised controllers (ASO, PSO). Based on the DC bus voltage results, it can be determined that the proposed hybrid (ASO-PSO) tuned FOPI controllers of the BES system reach steady state in less time (0.039s)

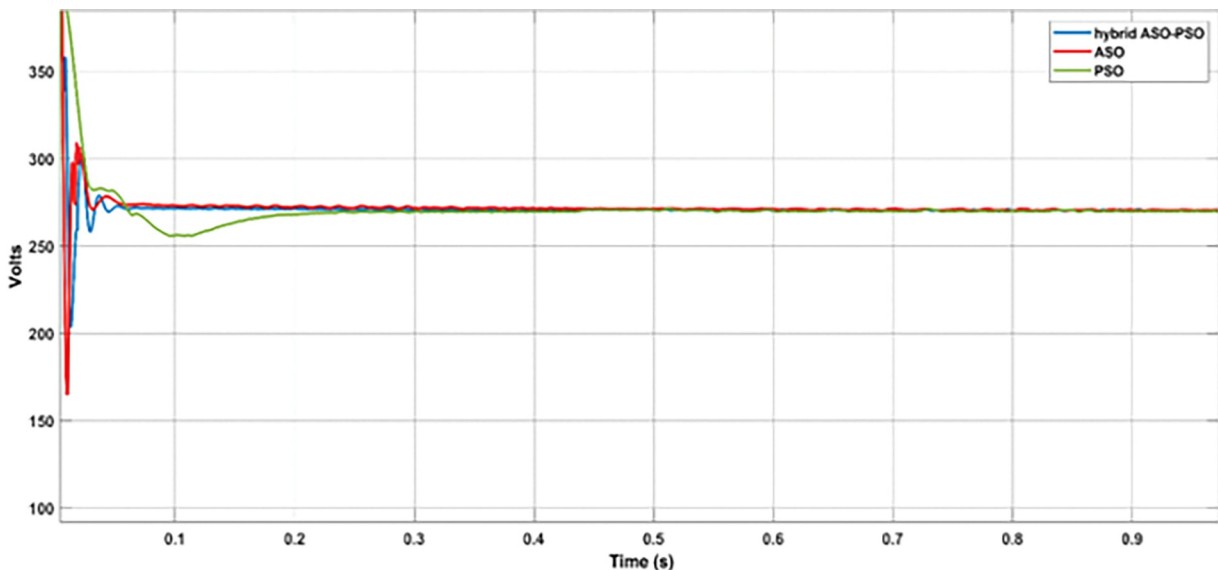

**Fig 17. Results of settling time with proposed controller.**

**Table 2. Results of control response (FOPI controllers with different techniques).**

| Optimized Methods | Settling time (s) | Peak Overshoot (%) | Rise time (s) |
|---|---|---|---|
| Hybrid ASO-PSO | 0.039 | 45.8% | 0.0018 |
| ASO | 0.049 | 47.2% | 0.0019 |
| PSO | 0.159 | 48% | 0.002 |

with less overshoot (45.8%) than the ASO (0.049s and 47.2%) and PSO (0.159s and 48%) tuned controllers. Likewise, improved rise time (0.0018 s) was observed for the proposed hybrid controller than rise time of ASO (0.0019 s) and PSO (0.002 s) controllers. Thus, in comparison to other controllers, the proposed hybrid FOPI performs significantly better in terms of overshoot, settling time, and rise time.

**7.3.2 Analysis with Real time varying irradiance of solar PV.** In this analysis, the effect of real-time PV power variation in the MG network was considered to verify the dynamic response of the proposed hybrid BES controllers. To investigate this, real-time varying solar irradiance data from [36] was used. The real time solar irradiance data with duration of 10 s (Fig 18), was considered for this analysis. The varying solar profiles with corresponding variation in PV power are shown in Fig 19A and 19B, respectively.

According to the solar profile, maximum PV power of 8.1 kW and 7.3 kW was generated with solar irradiance of 870 Watts/m$^2$ (at 1.2 s) and 780 Watts/m$^2$ (at 6 s), respectively. During these times, the BES was permitted to charge the excess power (-1460 kW at 1.2 s and– 0.5 kW at 6 s). The BES, on the other hand, was permitted to discharge power of approximately 5 kW and 4.8 kW during low solar of 505 Watts/m$^2$ (at 2 s) and 495 Watts/m$^2$ (at 8 s), respectively. The charging and discharging cycle operation of BES, during the period of varying solar profile is shown in Fig 20. From the results of BES power contribution, it can be noticed that the proposed hybrid (PSO-ASO) FOPI controllers of BES system attains steady state power level more quickly than other controllers (ASO and PSO) during the charge and discharge cycle of operation. Fig 20 depicts the charge and discharge cycle operation of the BES during the real time

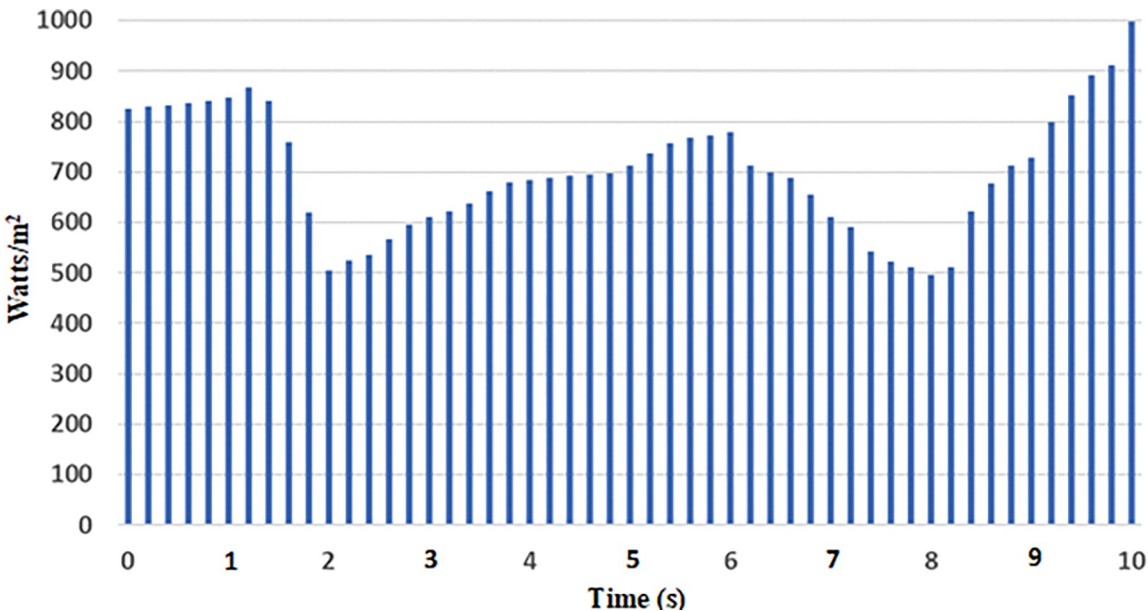

**Fig 18. Profile of Solar Irradiance Data.**

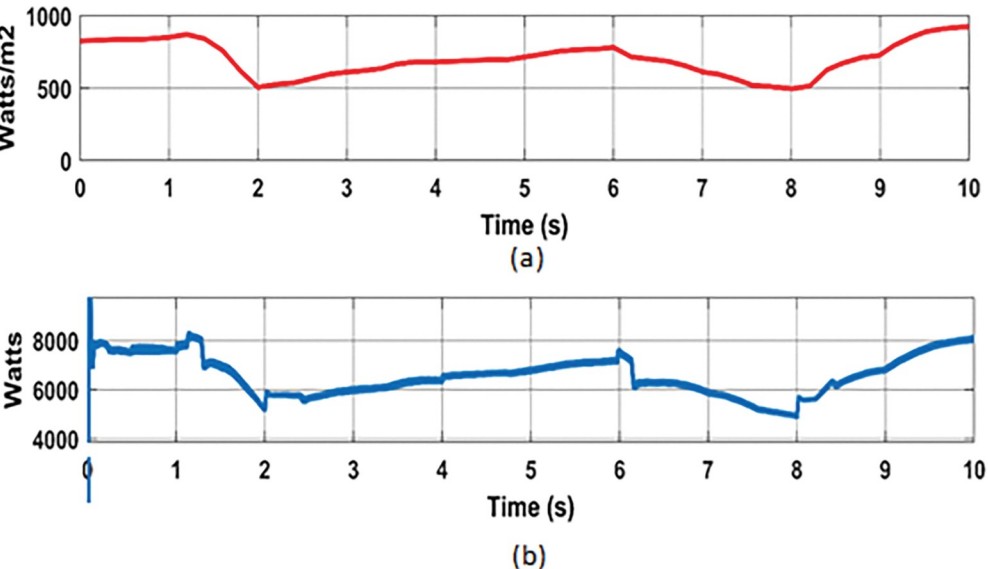

**Fig 19.** Solar PV System: (a) Solar irradiance; (b) PV power.

varying solar profile of PV system. As can be seen from the findings of the BES power contribution, the hybrid (PSO-ASO) FOPI controllers of the BES system reach steady state power level more quickly than other controllers (ASO and PSO) during the charge and discharge cycle of operation. Thus, the proposed hybrid controllers of BES system are more robust in terms of steady state control response during the real time variation of PV power in MG network.

Fig 21 depicts the results of DC bus voltage deviations (zoomed view) observed throughout the simulation analysis of the entire solar profile. According to the results, the proposed hybrid controllers of BES system have a significant reduction in voltage deviations and require less time to achieve a steady state value of DC bus voltage than the ASO and PSO controllers.

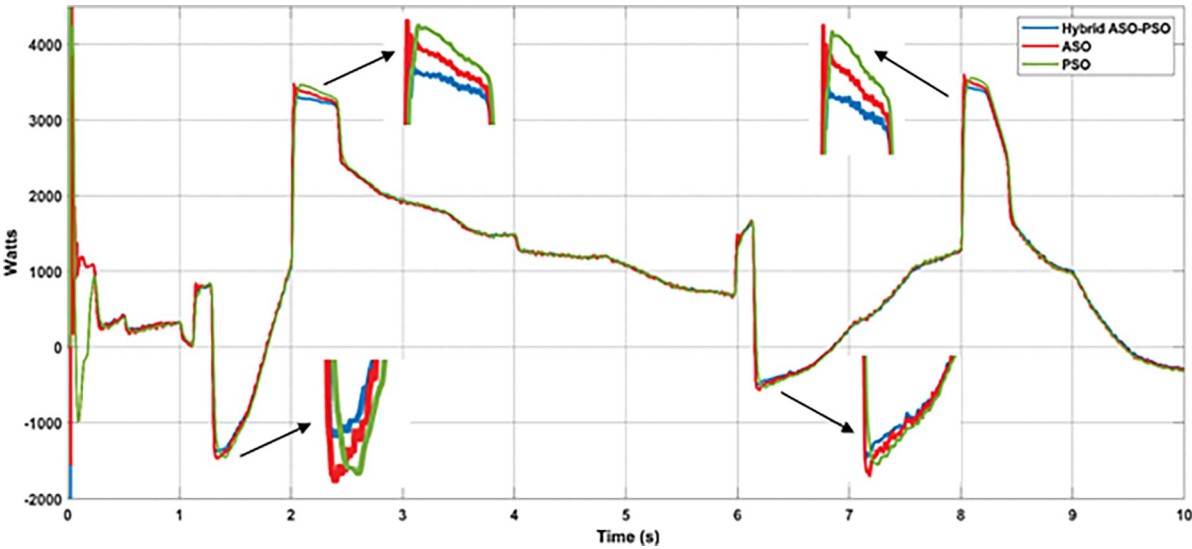

**Fig 20. Results of BES power under the real time uncertainty of solar PV.**

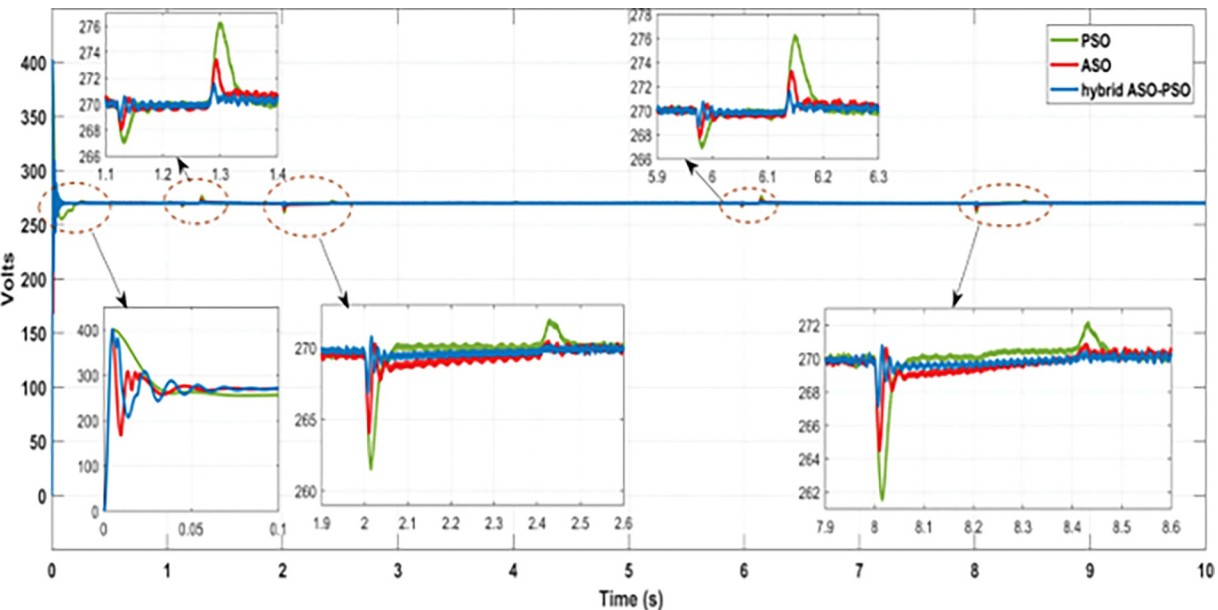

**Fig 21. Results of DC bus voltage deviations under the real time uncertainty of solar PV.**

Fig 22 and Table 3 illustrates the level of voltage deviations, observed during maximum and minimum solar conditions.

With respect to the reference bus voltage (270 V) of DC MG, a significant reduction of voltage deviations were observed with the proposed hybrid controllers of BES at maximum (0.55% with 870 Watts/m$^2$ and 0.44% with 780 Watts/m$^2$) solar irradiance of PV system than the ASO (1.1% with 870 Watts/m$^2$ and 0.93% with 780 Watts/m$^2$) and PSO (2.2% with 870 Watts/m$^2$ and 2.07% with 780 Watts/m$^2$) controllers. Likewise, reduced voltage deviations were observed with proposed controllers of BES at minimum (1.1% with 870 Watts/m$^2$ and 1.7% with 780

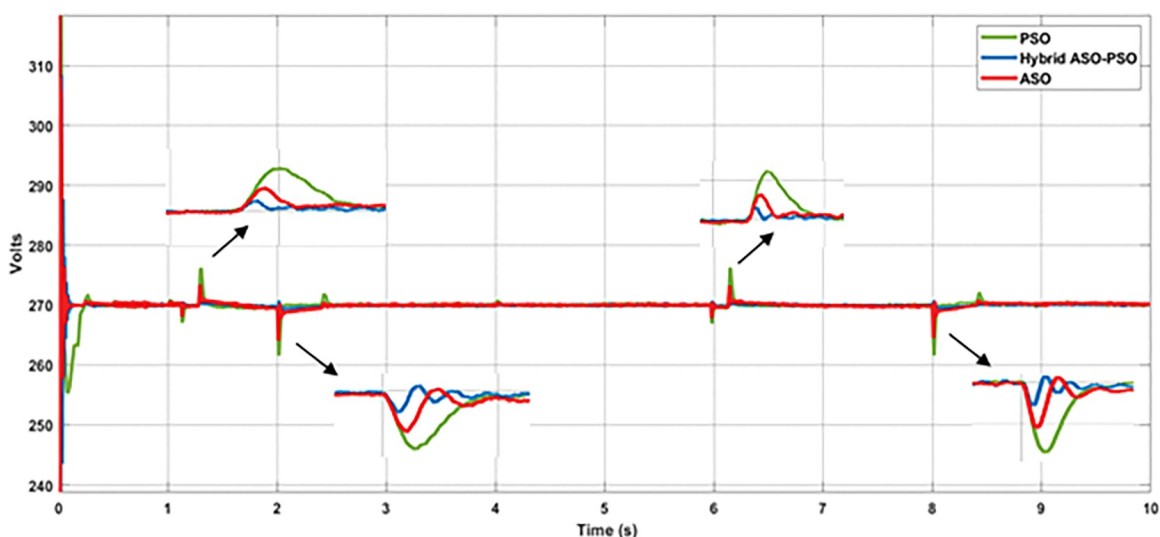

**Fig 22. Results of voltage deviations under the maximum and minimum solar conditions.**

**Table 3. Results of voltage deviations during maximum and minimum solar conditions.**

| Solar Conditions | Optimization methods | | |
|---|---|---|---|
| | Hybrid ASO-PSO | ASO | PSO |
| Max. Solar 870 Watts/m$^2$ | 271.5 V | 273 V | 276 V |
| Voltage deviations | 0.55% | 1.11% | 2.22% |
| Min. Solar 505 Watts/m$^2$ | 267 V | 264.3 V | V |
| Voltage deviations | - 1.11% | - 2.11% | - 3.15% |
| Max. Solar 780 Watts/m$^2$ | 271.2 V | 272.5 V | 275.6 V |
| Voltage deviations | 0.44% | 0.92% | 2.07% |
| Min. Solar 495 Watts/m$^2$ | 265.5 V | 264.6 V | 261.8 |
| Voltage deviations | - 1.67% | - 2% | - 3.04% |

Watts/m$^2$) solar irradiance than the ASO (2.1% with 870 Watts/m$^2$ and 2% with 780 Watts/m$^2$) and PSO (3.1% with 870 Watts/m$^2$ and 3.03% with 780 Watts/m$^2$) controllers. Thus, when compared to ASO and PSO controllers, the proposed hybrid controllers of BES offer better control in voltage deviations of more than 0.5% with ASO and 1% with PSO and more than 1% with ASO and 2% with PSO during maximum and minimum solar conditions, respectively.

According to the overall analysis results, the proposed FOPI controller of the BES system is more robust in terms of control response and maintaining minimal voltage deviations during the random and real time dynamic conditions of MG network

**7.3.3 Performance of convergence.** The convergence performance of proposed ASO-PSO control algorithm was verified in terms of minimizing the steady state performance indices of the IAE objective function, as given in Eq (4). From the results of fitness curves as shown in Fig 23, it can be determined that the proposed ASO-PSO control method converges faster in lesser iterations (at 42) than other ASO and PSO control methods. Thus, the proposed ASO-PSO control method outperforms in terms of providing a better solution and balanced trade-off between exploration and exploitation to prevent confinement into local minima.

## Conclusions

In this paper, a hybrid RE integrated DC MG model is developed in Matlab-Simulink software environment with implementation of BES based PMCS to get better power balance and voltage regulation under the uncertain conditions of RE sources and varying load profile in DC network. Additionally, using a hybrid ASO-PSO optimization technique, a BMS control with optimised FOPI controllers is proposed for the BES system. Under dynamic conditions like changing load profiles and uncertainty of RE sources, the proposed hybrid controller of the BES system enhances control response and keeps voltage deviations to a minimum (PV and wind turbine) in DC network. The PMCS with BMS control ensures safe operation of BES with better voltage regulation in the MG network, and the PMCS with source control improves the reliability of the MG system in the event of insufficient power generation from RE sources and depleted battery storage. In addition, the hybrid ASO-PSO tuned FOPI controllers of the BES system offer superior control performance (in terms of settling time, rise time, and peak overshoot) when compared to FOPI controllers of the ASO and PSO methods during the random variation of RE sources and connected load in the MG network. Furthermore, the proposed hybrid controllers of BES system outperforms other control methods in terms of steady

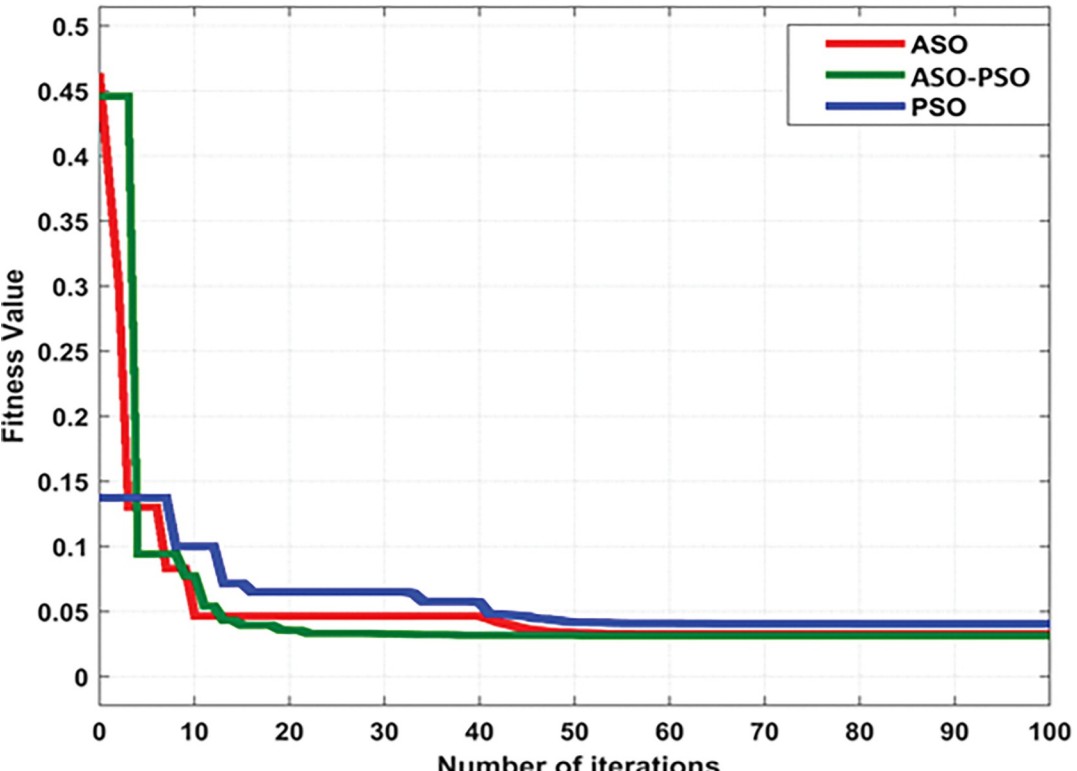

**Fig 23. Convergence curves of optimized control methods.**

state control response and maintaining minimal voltage deviations even at real time varying solar irradiance of PV system. The future scope of this research will include the implementation of chaotic search-based hybrid optimization algorithms for optimizing the BES controllers.

## Supporting information

**S1 File. Codes data.**
(ZIP)

## Acknowledgments

The authors gratefully acknowledge the Advanced Lightning, Power and Energy Research (ALPER), Universiti Putra for providing facilities to carry out the research.

## Author Contributions

**Conceptualization:** Khaizaran Al sumarmad, Nasri Sulaiman.

**Data curation:** Khaizaran Al sumarmad.

**Formal analysis:** Khaizaran Al sumarmad.

**Investigation:** Khaizaran Al sumarmad, Nasri Sulaiman.

**Methodology:** Khaizaran Al sumarmad, Nasri Sulaiman.

**Project administration:** Noor Izzri Abdul Wahab, Hashim Hizam.

**Resources:** Khaizaran Al sumarmad.

**Software:** Khaizaran Al sumarmad.

**Supervision:** Nasri Sulaiman, Noor Izzri Abdul Wahab, Hashim Hizam.

**Validation:** Khaizaran Al sumarmad.

**Visualization:** Khaizaran Al sumarmad, Nasri Sulaiman.

**Writing – original draft:** Khaizaran Al sumarmad.

**Writing – review & editing:** Khaizaran Al sumarmad.

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
