## [Decision Letter · Decision Letter 0]

14 Apr 2023

PONE-D-23-08007Implementation of hybrid optimized Battery controller and Advanced Power Management Control Strategy in a Renewable Energy Integrated DC MicrogridPLOS ONE

Dear Dr. Al sumarmad,

Thank you for submitting your manuscript to PLOS ONE. After careful consideration, we feel that it has merit but does not fully meet PLOS ONE’s publication criteria as it currently stands. Therefore, we invite you to submit a revised version of the manuscript that addresses the points raised during the review process.

We look forward to receiving your revised manuscript.

Kind regards,

Yogendra Arya

Academic Editor

PLOS ONE

Journal Requirements:

Reviewers' comments:

Reviewer's Responses to Questions

**Comments to the Author**

1. Is the manuscript technically sound, and do the data support the conclusions?

Reviewer #1: Yes

Reviewer #2: Yes

2. Has the statistical analysis been performed appropriately and rigorously? 

Reviewer #1: Yes

Reviewer #2: Yes

3. Have the authors made all data underlying the findings in their manuscript fully available?

Reviewer #1: Yes

Reviewer #2: Yes

4. Is the manuscript presented in an intelligible fashion and written in standard English?

Reviewer #1: Yes

Reviewer #2: Yes

5. Review Comments to the Author

Reviewer #1: The title of the paper is good and the manuscript has written with well organized. Please consider to address the following review comments:

1. Author needs to mention whether the PMCS with BES control is implemented for Islanded (Autonomous) mode of MG operation, if it is developed for islanded mode, state the reason for considering this mode

2.Author need to mention the tuning mode, whether it is on line or off line mode, and mention the configuration of computer used for tuning that

3. In equation 17, states Ploss, what it includes (expand the expression of Ploss)

4. In PMCS with source control analysis, once the battery SOC level reaches to 30 %, load management switches off non-essential loads and SOC reaches further below 30 %, battery went idle mode and grid breaker on. Once grid source on, battery starts charging back, the questuion is, on what SOC level, the non essential loads going to switch on back ?

5. In addition to the results discussion, it is required to add % of voltage deviations for each method in Table1 and Table3

6. Mention, at what number of iterations, the proposed ASO-PSO method gets converged

7. Check grammatical and spell check errors through out the manuscript

Reviewer #2: Figure 9 PMCS analysis with BMS control: (a) Load power; (b) Battery power; (c) Battery SOC; (d) DC bus voltage  Power balancing can be shown through plot as PMCS is meant for satisfying this power balancing condition (total generation =total demand+ losses)

Figure 14 Results of voltage deviations at different events & Table 3 Results of voltage deviations during maximum and minimum solar conditions  can be expressed in terms of percentage of deviation

Table 2 Results of control response (FOPI controllers with different techniques) - Peak Overshoot (%) is not upto the standard values. Hence the controller needs to be revised with Differential controller to minimize the overshoot.

6. PLOS authors have the option to publish the peer review history of their article (what does this mean?). If published, this will include your full peer review and any attached files.

Reviewer #1: No

Reviewer #2: No

---

## [Author Response · Author response to Decision Letter 0]

14 May 2023

Dear Editor& Reviewer(s),

Thank you very much for your valuable feedback and time in reviewing the article. All your feedback helped us to improve the quality of the manuscript. The answers for the reviewer comments are listed as follows;

Reviewer #1:

Question 1: Author needs to mention whether the PMCS with BES control is implemented for Islanded (Autonomous) mode of MG operation, if it is developed for islanded mode, state the reason for considering this mode

Answer: Thanks for your valuable feedback. The utilization of PMCS with BMS control has been proposed for islanded MG network. The absence of grid support, limited fault current, restricted generation capacity, and variable power output from RE sources in islanded MG network pose significant obstacles in achieving adequate control and voltage regulation during transient scenarios, compared to grid-connected MG. Therefore, to ensure reliable operation of MG in islanded mode, it is given priority to implement an advanced PMCS with BMS control for islanded MG network than grid connected MG.

Question 2: Author need to mention the tuning mode, whether it is on line or off line mode, and mention the configuration of computer used for tuning that 

Answer: Thanks for your comments. The optimization is considered in off line while tuning the FOPI controllers of battery converter. The configuration of personal computer “Intel (R) core (TM) i5-2410 M CPU @2.30 GHz/ 16 GB RAM” is used for tuning process. 

Question 3: In equation 17, states Ploss, what it includes (expand the expression of Ploss)

Answer: Thanks for your feedback. PLOSS is generally expressed as product of network resistance (R) and square of the current (I2) flowing through the DC network and same is updated in the manuscript.

Question 4: In PMCS with source control analysis, once the battery SOC level reaches to 30 %, load management switches off non-essential loads and SOC reaches further below 30 %, battery went idle mode and grid breaker on. Once grid source on, battery starts charging back, the question is, on what SOC level, the non-essential loads going to switch on back?

Answer: Thanks for your comments. Whenever SOC level of battery reaches above 50 %, the non-essential loads will be switched on back, only with fulfilment of any one of the following conditions: either grid source is on or PV power generation is at maximum level.

Question 5: In addition to the results discussion, it is required to add % of voltage deviations for each method in Table1 and Table3

Answer: Thanks for your valuable feedback. In Table 1 and Table 3, the % of voltage deviations are included.

Question 6: Mention, at what number of iterations, the proposed ASO-PSO method gets converged

Answer: Thanks for your feedback. The proposed ASO-PSO method gets converged at the 42nd iterations (Figure 22) and same updated in the section 7.3.3 of manuscript.

Question 7: Check grammatical and spell check errors throughout the manuscript

Answer: Answer: Thanks for your comments. The spell check and grammatical errors checked throughout the manuscript. 

Reviewer #2:

Question 1: Figure 9 PMCS analysis with BMS control: (a) Load power; (b) Battery power; (c) Battery SOC; (d) DC bus voltage  Power balancing can be shown through plot as PMCS is meant for satisfying this power balancing condition (total generation =total demand+ losses)

Answer: Thanks for your valuable suggestion. The power balance among sources and load are shown in a plot. The power balance with align to the battery charging/discharging power (maximum/minimum) during the cases of varying solar, wind, and load power are shown in Figure 10 and same is updated in the manuscript.

Question 2: Figure 14 Results of voltage deviations at different events & Table 3 Results of voltage deviations during maximum and minimum solar conditions  can be expressed in terms of percentage of deviation

Answer: Thanks for your valuable feedback. The % of voltage deviations are updated in the Table 1 and Table 3 of manuscript.

Question 3: Table 2 Results of control response (FOPI controllers with different techniques) - Peak Overshoot (%) is not upto the standard values. Hence the controller needs to be revised with Differential controller to minimize the overshoot.

Answer: Thanks for your valuable input. To mitigate the issue of peak overshoot, we intended to utilize an advanced robust controller in the future stages of this study.

---

## [Decision Letter · Decision Letter 1]

31 May 2023

Implementation of hybrid optimized Battery controller and Advanced Power Management Control Strategy in a Renewable Energy Integrated DC Microgrid

PONE-D-23-08007R1

Dear Dr. Al sumarmad,

We’re pleased to inform you that your manuscript has been judged scientifically suitable for publication and will be formally accepted for publication once it meets all outstanding technical requirements.

Kind regards,

Yogendra Arya

Academic Editor

PLOS ONE

Additional Editor Comments (optional):

Reviewers' comments:

Reviewer's Responses to Questions

**Comments to the Author**

1. If the authors have adequately addressed your comments raised in a previous round of review and you feel that this manuscript is now acceptable for publication, you may indicate that here to bypass the “Comments to the Author” section, enter your conflict of interest statement in the “Confidential to Editor” section, and submit your "Accept" recommendation.

Reviewer #1: All comments have been addressed

Reviewer #2: All comments have been addressed

2. Is the manuscript technically sound, and do the data support the conclusions?

Reviewer #1: Yes

Reviewer #2: Yes

3. Has the statistical analysis been performed appropriately and rigorously? 

Reviewer #1: Yes

Reviewer #2: Yes

4. Have the authors made all data underlying the findings in their manuscript fully available?

Reviewer #1: Yes

Reviewer #2: Yes

5. Is the manuscript presented in an intelligible fashion and written in standard English?

Reviewer #1: Yes

Reviewer #2: Yes

6. Review Comments to the Author

Reviewer #1: Thanks for addressing all the comments. All the points ar successfully addressed by the authors with full of satisfaction.

Reviewer #2: (No Response)

7. PLOS authors have the option to publish the peer review history of their article (what does this mean?). If published, this will include your full peer review and any attached files.

Reviewer #1: No

Reviewer #2: **Yes: **RAJESWARI RAMACHANDRAN

---

## [Editor Report · Acceptance letter]

5 Jun 2023

PONE-D-23-08007R1 

Implementation of hybrid optimized Battery controller and Advanced Power Management Control Strategy in a Renewable Energy Integrated DC Microgrid 

Dear Dr. Al sumarmad:

I'm pleased to inform you that your manuscript has been deemed suitable for publication in PLOS ONE. Congratulations! Your manuscript is now with our production department. 

Kind regards, 

on behalf of

Dr. Yogendra Arya 

Academic Editor

PLOS ONE